# Explainable Brain Age Prediction using coVariance Neural Networks

**Saurabh Sihag**[†]**, Gonzalo Mateos**[‡]**, Corey McMillan**[†]**, Alejandro Ribeiro**[†]

[†]University of Pennsylvania, Philadelphia, PA.
[‡]University of Rochester, Rochester, NY.

## Abstract

In computational neuroscience, there has been an increased interest in developing machine learning algorithms that leverage brain imaging data to provide estimates of "brain age" for an individual. Importantly, the discordance between brain age and chronological age (referred to as "brain age gap") can capture accelerated aging due to adverse health conditions and therefore, can reflect increased vulnerability towards neurological disease or cognitive impairments. However, widespread adoption of brain age for clinical decision support has been hindered due to lack of transparency and methodological justifications in most existing brain age prediction algorithms. In this paper, we leverage coVariance neural networks (VNN) to propose an explanation-driven and anatomically interpretable framework for brain age prediction using cortical thickness features. Specifically, our brain age prediction framework extends beyond the coarse metric of brain age gap in Alzheimer's disease (AD) and we make two important observations: (i) VNNs can assign anatomical interpretability to elevated brain age gap in AD by identifying contributing brain regions, (ii) the interpretability offered by VNNs is contingent on their ability to exploit specific eigenvectors of the anatomical covariance matrix. Together, these observations facilitate an explainable and anatomically interpretable perspective to the task of brain age prediction.

## 1 Introduction

Aging is characterized by progressive changes in the anatomy and function of the brain [1] that can be captured by different modalities of neuroimaging [2, 3]. Importantly, individuals can age at variable rates, a phenomenon described as "biological aging" [4]. Numerous existing studies based on a large spectrum of machine learning approaches study brain-predicted biological age, also referred to as brain age, which is derived from neuroimaging data [5–12]. Accelerated aging, i.e., when biological age is elevated as compared to chronological age (time since birth), may predict age-related vulnerabilities like risk for cognitive decline or neurological conditions like Alzheimer's disease (AD) [13, 14]. In this domain, the metric of interest is *brain age gap*, i.e., the difference between brain age and chronological age. We use the notation $\Delta$-Age to refer to the brain age gap.

Inferring $\Delta$-age from neuroimaging data presents a unique statistical challenge as it is essentially a qualitative metric with no ground truth and is expected to be elevated in individuals with underlying neurodegenerative condition as compared to the healthy population[12, 15]. The existing machine learning approaches for inferring $\Delta$-Age commonly rely on regression models trained to predict chronological age for a healthy population. Under the hypothesis that such models can detect accelerated aging, they are applied to cohorts representing adverse health conditions. From a statistical perspective, the residuals of the regression models inform the $\Delta$-Age estimates with the

Code: https://github.com/sihags/VNN_Brain_Age

expectation that they will degrade in a specific direction when deployed to predict chronological age for individuals with adverse health conditions. Hence, it is paramount to analyze the structure and statistics of the residuals of the model to validate whether $\Delta$-Age inferred using them provide biologically plausible information about the adverse health condition. A layman overview of the procedure of inferring $\Delta$-Age is included in Appendix D. In this paper, we focus on brain age prediction using cortical thickness features. Cortical thickness evolves with normal aging [16] and is impacted due to neurodegeneration [17, 18]. Further, the age-related and disease severity related variations also appear in anatomical covariance matrices evaluated from the cortical thickness [19].

**Existing literature.** The current state-of-the-art deep learning methods in the brain age prediction domain focus exclusively on the performance of the model on predicting chronological age for a healthy population as a metric for assessing the quality of their approach[20–22]. We refer to such methods as *performance-driven approaches* to brain age prediction. Major criticisms of such performance-driven approaches include the coarseness of $\Delta$-Age that results in lack of specificity of brain regions contributing to the elevated $\Delta$-Age; and the lack of clarity regarding the reliance on the prediction accuracy for chronological age in the design of these brain age prediction models [5, 23].

To address the criticism regarding the lack of interpretability or explainability of $\Delta$-Age, recent studies have utilized state-of-the-art post-hoc, model-agnostic methods, such as, SHAP, LIME [24], saliency maps [20, 25], and layer-wise relevance propagation [26] in conjunction with the performance-driven approaches. These methods commonly add anatomical interpretability to brain age estimates by assigning importance to the input features (often associated with specific anatomic regions). However, interpretability offered by such post-hoc approaches may not be conclusive if not shown to be stable to small perturbations to the input, variations in training algorithms and model multiplicity (i.e., when multiple models with similar performance may exist but offer distinct explanations) [27–29].

There exists sparse empirical evidence in the existing literature that hints at decoupling the task of brain age prediction from the performance achieved by the model in predicting chronological age for healthy population. For instance, a previous study has reported that models with a 'moderate' performance for predicting chronological age achieved a more informative brain age [21]. However, an appropriate 'moderate' fit on the chronological age that leads to the most informative brain age may not be generalizable to diverse datasets (diverse in terms of sample sizes, for example). Furthermore, a recent study of several existing brain age prediction frameworks has revealed that the accuracy achieved on the chronological age prediction task may not correlate with the clinical utility of associated $\Delta$-Age estimates [30]. Intuitively, the performance on chronological age prediction task is an incomplete, if not flawed, metric for assessing the quality of $\Delta$-Age estimate, as it cannot readily provide clarity on the correlation between the performance on predicting chronological age for healthy population and clinical utility of $\Delta$-Age.

**Explainable perspective to brain age prediction.** In this paper, we propose a principled framework for brain age prediction based on the recently studied coVariance neural networks (VNNs) [31]. VNN is a graph neural network (GNN) that operates on the sample covariance matrix as the graph and achieves learning objectives by manipulating the input data according to the eigenvectors (or principal components) of the covariance matrix. Thus, VNNs are inherently explainable models, as their inference outcomes can be linked with their ability to exploit the eigenvectors of the covariance matrix. In this context, the explainability offered by VNNs can be categorized as model-level explainability according to the taxonomy of explainability methods discussed in [32]. In general, model-level explainability can offer a more fundamental and generic understanding of the model than the aforementioned explainability methods applied in the brain age prediction application (such methods can broadly be categorized as instance-level methods [32]). A survey of explainability methods in GNNs is provided in Appendix B.

For the task of brain age prediction, the key focus of this paper is not on the accuracy in predicting chronological age, but rather *(i) what properties does a VNN gain when it is exposed to the information provided by chronological age of healthy controls,* and *(ii) whether and how these properties could translate to a meaningful brain age estimate.* While highly relevant, these aspects are often overlooked in existing studies on brain age prediction. In this context, VNNs provide novel insights beyond that possible by focusing only on model performance. Specifically, training VNNs to predict chronological age using cortical thickness features from the healthy population fine-tuned their ability to exploit the eigenvectors of the anatomical covariance matrix. Further, the statistical analyses of the outputs of the final layer of the VNN allowed us to identify the most significant contributors to elevated $\Delta$-Age

in AD with respect to healthy population. Mapping these contributors on the brain surface rendered an anatomically interpretable perspective to $\Delta$-Age estimates. Finally, the anatomical interpretability offered by VNNs to $\Delta$-Age prediction in AD was strongly associated with certain eigenvectors of the covariance matrix, thus, rendering an explainable perspective to brain age in terms of the ability of VNNs to exploit the eigenvectors of the covariance matrix in a specific manner. We emphasize here that in this paper, the term 'interpretability' is used in the context of anatomic interpretability of $\Delta$-Age and the term 'explainability' refers to explaining the VNN inference outcomes in terms of their associations with the eigenvectors of the covariance matrix.

**Contributions.** The contributions in this paper can be summarized as follows.

a) **VNNs provide anatomically interpretable $\Delta$-Age:** $\Delta$-Age in individuals with AD diagnosis was elevated as compared to healthy controls and significantly correlated with a clinical marker of dementia severity. Moreover, by analyzing the outputs at the final layer of VNN for AD and healthy population and mapping the results on the brain surface, we could identify contributing brain regions to elevated $\Delta$-Age in AD. Hence, VNN architecture yielded anatomical interpretability for $\Delta$-Age (Fig. 2).

b) **Anatomical interpretability correlated with eigenvectors of the anatomical covariance matrix:** Our experiments demonstrated that certain eigenvectors of the anatomical covariance matrix were highly correlated with the features that facilitated anatomical interpretability for $\Delta$-Age (Fig. 3). Thus, learning to predict chronological age of healthy population facilitated the ability of VNNs to exploit the eigenvectors of the anatomical covariance matrix that were relevant to $\Delta$-Age, yielding an explainable perspective to brain age prediction.

We focused our analysis on open access OASIS-3 dataset consisting of cortical thickness features from cognitively normal individuals and individuals in various stages of cognitive decline [33]. The findings were further validated on the baseline ADNI-1 dataset [34]. The utility of VNNs in predicting $\Delta$-Age has been explored previously in [35] but no insights were provided regarding their explainability or anatomical interpretability. See Appendix B for other relevant studies.

## 2 coVariance Neural Networks

We begin with a brief introduction to VNNs. VNNs inherit the architecture of GNNs [36] and operate on the sample covariance matrix as the graph [31]. A dataset consisting of $n$ random, independent and identically distributed (i.i.d) samples, given by $\mathbf{x}_i \in \mathbb{R}^{m \times 1}, \forall i \in \{1, \ldots, n\}$, can be represented in matrix form as $\mathbf{X}_n = [\mathbf{x}_1, \ldots, \mathbf{x}_n]$. Using $\mathbf{X}_n$, the sample covariance matrix is estimated as

$$\mathbf{C} \triangleq \frac{1}{n-1} \sum_{i=1}^{n} (\mathbf{x}_i - \bar{\mathbf{x}})(\mathbf{x}_i - \bar{\mathbf{x}})^{\mathsf{T}} , \tag{1}$$

where $\bar{\mathbf{x}}$ is the sample mean of samples in $\mathbf{X}_n$. The covariance matrix $\mathbf{C}$ can be viewed as the adjacency matrix of a graph representing the stochastic structure of the dataset $\mathbf{X}_n$, where the $m$ dimensions of the data can be thought of as the nodes of an $m$-node, undirected graph and its edges represent the pairwise covariances between different dimensions.

### 2.1 Architecture

Similar to GNNs that rely on convolution operations modeled by *linear-shift-and-sum* operators [36, 37], the convolution operation in a VNN is modeled by a coVariance filter, given by

$$\mathbf{H}(\mathbf{C}) \triangleq \sum_{k=0}^{K} h_k \mathbf{C}^k , \tag{2}$$

where scalar parameters $\{h_k\}_{k=0}^{K}$ are referred to as filter taps that are learned from the data. The application of coVariance filter $\mathbf{H}(\mathbf{C})$ on an input $\mathbf{x}$ translates to combining information across different sized neighborhoods. For $K > 1$, the convolution operation combines information across multi-hop neighborhoods (up to $K$-hop) according to the weights $h_k$ to form the output $\mathbf{z} = \mathbf{H}(\mathbf{C})\mathbf{x}$.

A single layer of VNN is formed by passing the output of the coVariance filter through a non-linear activation function $\sigma(\cdot)$ (e.g., ReLU, $\tanh$) that satisfies $\sigma(\mathbf{u}) = [\sigma(u_1), \ldots, \sigma(u_m)]$ for $\mathbf{u} = [u_1, \ldots, u_m]$. Hence, the output of a single layer VNN with input $\mathbf{x}$ is given by $\mathbf{z} = \sigma(\mathbf{H}(\mathbf{C})\mathbf{x})$. The construction of a multi-layer VNN is formalized next.

**Remark 1** (Multi-layer VNN). *For an L-layer VNN, denote the coVariance filter in layer $\ell$ of the VNN by $\mathbf{H}_\ell(\mathbf{C})$ and its corresponding set of filter taps by $\mathcal{H}_\ell$. Given a pointwise nonlinear activation function $\sigma(\cdot)$, the relationship between the input $\mathbf{x}_{\ell-1}$ and the output $\mathbf{x}_\ell$ for the $\ell$-th layer is*

$$\mathbf{x}_\ell = \sigma(\mathbf{H}_\ell(\mathbf{C})\mathbf{x}_{\ell-1}) \quad for \quad \ell \in \{1, \ldots, L\} , \tag{3}$$

*where $\mathbf{x}_0$ is the input $\mathbf{x}$.*

Furthermore, similar to other deep learning models, sufficient expressive power can be facilitated in the VNN architecture by incorporating multiple input multiple output (MIMO) processing at every layer. Formally, consider a VNN layer $\ell$ that can process $F_{\ell-1}$ number of $m$-dimensional inputs and outputs $F_\ell$ number of $m$-dimensional outputs via $F_{\ell-1} \times F_\ell$ number of filter banks [38]. In this scenario, the input is specified as $\mathbf{X}_{\text{in}} = [\mathbf{x}_{\text{in}}[1], \ldots, \mathbf{x}_{\text{in}}[F_{\text{in}}]]$, and the output is specified as $\mathbf{X}_{\text{out}} = [\mathbf{x}_{\text{out}}[1], \ldots, \mathbf{x}_{\text{out}}[F_{\text{out}}]]$. The relationship between the $f$-th output $\mathbf{x}_{\text{out}}[f]$ and the input $\mathbf{x}_{\text{in}}$ is given by $\mathbf{x}_{\text{out}}[f] = \sigma\left(\sum_{g=1}^{F_{\text{in}}} \mathbf{H}_{fg}(\mathbf{C})\mathbf{x}_{\text{in}}[g]\right)$, where $\mathbf{H}_{fg}(\mathbf{C})$ is the coVariance filter that processes $\mathbf{x}_{\text{in}}[g]$. Without loss of generality, we assume that $F_\ell = F, \forall \ell \in \{1, \ldots, L\}$. In this case, the set of all filter taps is given by $\mathcal{H} = \{\mathcal{H}_{fg}^\ell\}, \forall f, g \in \{1, \ldots, F\}, \ell \in \{1, \ldots, L\}$, where $\mathcal{H}_{fg} = \{h_{fg}^\ell[k]\}_{k=0}^K$ and $h_{fg}^\ell[k]$ is the $k$-th filter tap for filter $\mathbf{H}_{fg}(\mathbf{C})$. Thus, we can compactly represent a multi-layer VNN architecture capable of MIMO processing via the notation $\Phi(\mathbf{x}; \mathbf{C}, \mathcal{H})$, where the set of filter taps $\mathcal{H}$ captures the full span of its architecture. We also use the notation $\Phi(\mathbf{x}; \mathbf{C}, \mathcal{H})$ to denote the output at the final layer of the VNN. Various aspects of the VNN architecture are illustrated in Fig. 6 in Appendix E. The VNN final layer output $\Phi(\mathbf{x}; \mathbf{C}, \mathcal{H})$ is succeeded by a readout function that maps it to the desired inference outcome.

**Remark 2** (Statistical inference using VNNs). *Given the eigendecomposition of $\mathbf{C} = \mathbf{V}\mathbf{\Lambda}\mathbf{V}^\mathsf{T}$, the spectral properties of the VNN are established by studying the projection of the coVariance filter output $\mathbf{z} = \mathbf{H}(\mathbf{C})\mathbf{x}$ on the eigenvectors $\mathbf{V}$ (similar to that for a graph filter using graph Fourier transform [39, 40]). Theorem 1 in [31] established the equivalence between processing data samples with principal component analysis (PCA) transform and processing data samples with a specific polynomial on the covariance matrix $\mathbf{C}$. Hence, it can be concluded that input data is processed with VNNs, at least in part, by exploiting the eigenvectors of $\mathbf{C}$. Unlike simpler PCA-based inference models, VNNs offer stability [31] and transferability guarantees [35], which ensure reproducibility of the inference outcomes by VNNs with high confidence.*

In the context of brain age prediction, we leverage the observations in Remark 2 to demonstrate the relationships of the inference outcomes with the eigenvectors of the anatomical covariance matrix $\mathbf{C}$ (estimated from cortical thickness features) in Section 4.

## 2.2 VNN Learning

The VNN model is trained for a regression task, where the chronological age is predicted using $m$ cortical thickness features. Since the VNN architecture has $F$ number of $m$-dimensional outputs in the final layer, $\Phi(\mathbf{x}; \mathbf{C}, \mathcal{H})$ is of dimensionality $m \times F$. The regression output is determined by a readout layer, which evaluates an unweighted mean of all outputs at the final layer of VNN. Therefore, the regression output for an individual with cortical thickness $\mathbf{x}$ is given by

$$\hat{y} = \frac{1}{Fm} \sum_{j=1}^m \sum_{f=1}^F [\Phi(\mathbf{x}; \mathbf{C}, \mathcal{H})]_{jf} . \tag{4}$$

Prediction using unweighted mean at the output implies that the VNN model exhibits permutation-invariance (i.e., the final output is independent of the permutation of the input features and covariance matrix). Moreover, it allows us to associate a scalar output with each brain region among the $m$ regions at the final layer. Specifically, we have

$$\mathbf{p} = \frac{1}{F} \sum_{f=1}^F [\Phi(\mathbf{x}; \mathbf{C}, \mathcal{H})]_f , \tag{5}$$

where $\mathbf{p}$ is the vector denoting the mean of filter outputs in the final layer's filter bank. Note that the mean of all elements in $\mathbf{p}$ is the prediction $\hat{y}$ formed in (4). In the context of cortical thickness

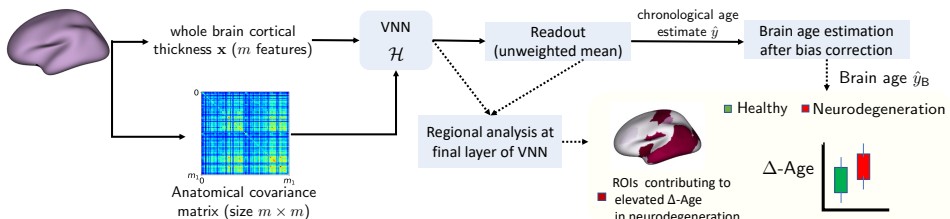

Figure 1: Workflow for brain age and $\Delta$-Age prediction using VNNs. Regions of interest (ROIs) contributing to elevated $\Delta$-Age in neurodegeneration were identified by mapping the results of the analyses of the outputs at the final layer of VNNs on the brain surface.

datasets, each element of $\mathbf{p}$ can be associated with a distinct brain region. Therefore, $\mathbf{p}$ is a vector of "regional contributions" to the output $\hat{y}$ by the VNN. This observation will be instrumental to establishing the interpretability offered by VNNs in the context of $\Delta$-Age prediction in Section 3. For a regression task, the training dataset $\{\mathbf{x}_i, y_i\}_{i=1}^n$ (where $\mathbf{x}_i$ are the cortical thickness data for an individual $i$ with chronological age $y_i$) is leveraged to learn the filter taps in $\mathcal{H}$ for the VNN $\Phi(\cdot; \mathbf{C}, \mathcal{H})$ such that they minimize the training loss, i.e.,

$$\mathcal{H}_{\text{opt}} = \min_{\mathcal{H}} \frac{1}{n} \sum_{i=1}^n \ell(\hat{y}_i, y_i) \,, \tag{6}$$

where $\hat{y}_i$ is evaluated similarly to (4) and $\ell(\cdot)$ is the mean-squared error (MSE) loss function.

## 3 Methods Overview for Brain Age Prediction

In this section, we provide an overview of the brain age prediction framework based on VNNs (see Fig. 1 for a summary). Our results primarily focus on the dataset described below.

**OASIS-3 Dataset**. This dataset was derived from publicly available freesurfer estimates of cortical thickness (derived from MRI images collected by 3 Tesla scanners and hosted on `central.xnat.org`), as previously reported [33], and comprised of cognitively normal individuals (HC; $n = 611$, age $= 68.38 \pm 7.62$ years, 351 females) and individuals with AD dementia diagnosis and at various stages of cognitive decline ($n = 194$, age $= 74.72 \pm 7.02$ years, 94 females). The cortical thickness features were curated according to the Destrieux (DKT) atlas [41](consisting of $m = 148$ cortical regions). For clarity of exposition of the brain age prediction method, any dementia staging to subdivide the group of individuals with AD dementia diagnosis into mild cognitive impairment (MCI) or AD was not performed, and we use the label AD+ to refer to this group. The individuals in AD+ group were significantly older than those in HC group (t-test: $p$-value $= 2.46 \times 10^{-23}$). The boxplots for the distributions of chronological age for HC and AD+ groups are included in Fig. 10 in Appendix J. For 191 individuals in the AD+ group, the clinical dementia rating (CDR) sum of boxes scores evaluated within one year (365 days) from the MRI scan were available (CDR sum of boxes $= 3.45 \pm 1.74$). CDR sum of boxes scores are commonly used in clinical settings to stage dementia severity [42] and were evaluated according to [43].

**Cross-validation**. The findings obtained via the analyses of the OASIS-3 dataset were cross-validated on the ADNI-1 dataset (described in Appendix I).

### 3.1 Training the VNNs on HC group

We first train the VNN model to predict chronological age using the cortical thickness features for the HC group. This enables the VNN models to capture the information about healthy aging from the cortical thickness and associated anatomical covariance matrix. The hyperparameters for the VNN architecture and learning rate of the optimizer were chosen according to a hyperparameter search procedure [44]. The VNN model had $L = 2$-layers with a filter bank, such that we had $F = 5$, and 6 filter taps in the first layer and 10 filter taps in the second layer. The learning rate for the Adam optimizer was set to 0.059. The number of learnable parameters for this VNN was 290. The HC group was split into a $90/10$ training/test split, and the covariance matrix was set to be the anatomical

covariance evaluated from the training set. A part of the training set was used as a validation set and the other used for training the VNN model. We trained 100 VNN models, each on a different permutation of the training data. The training process was similar for all VNNs and is described in Appendix F. No further training was performed for the VNN models in the subsequent analyses.

## 3.2 Analyses of regional residuals in AD+ and HC groups

Next, the VNN models trained to predict the chronological age for the HC group and (5) were adopted to study the effect of neurodegeneration on brain regions for AD+ group. Since the impact of neurodegeneration was expected to appear in the anatomical covariance matrix, we report the results when anatomical covariance $\mathbf{C}_{HA}$ from the combined cortical thickness data of HC and AD+ groups was deployed in the trained VNN models. Because of the stability property of VNNs [31, Theorem 3], the inference drawn from VNNs was expected to be stable to the composition of combined HC and AD+ groups used to estimate the anatomical covariance matrix $\mathbf{C}_{HA}$.

For every individual in the combined dataset of HC and AD+ groups, we processed their cortical thickness data $\mathbf{x}$ through the model $\Phi(\mathbf{x}; \mathbf{C}_{HA}, \mathcal{H})$ where parameters $\mathcal{H}$ were learned in the regression task on the data from HC group as described previously. Hence, the vector of mean of all final layer outputs for cortical thickness input $\mathbf{x}$ is given by $\mathbf{p} = \frac{1}{F} \sum_{f=1}^{F} [\Phi(\mathbf{x}; \mathbf{C}_{HA}, \mathcal{H}]_f$ and the VNN output is $\hat{y} = \frac{1}{148} \sum_{j=1}^{148} [\mathbf{p}]_j$. Furthermore, we define the residual for feature $a$ (or brain region represented by feature $a$ in this case) as

$$[\mathbf{r}]_a \triangleq [\mathbf{p}]_a - \hat{y} \ . \tag{7}$$

Thus, (7) allows us to characterize the residuals with respect to the VNN output $\hat{y}$ at the regional level, where brain regions are indexed by $a$. Henceforth, we refer to the residuals (7) as "regional residuals". Recall that these are evaluated for an individual with cortical thickness data $\mathbf{x}$. We hypothesized accelerated aging for an individual to be an aggregated effect of contributions from certain biologically plausible brain regions. The brain regions contributing to the observed higher $\Delta$-Age (procedure described in Section 3.3) could be characterized at a regional level by the analysis of regional residuals as defined in (7). Thus, the elements of the residual vector $\mathbf{r}$ can potentially act as a biomarker that can enable the isolation of brain regions impacted due to accelerated aging in AD.

In our experiments, for a given VNN model, the residual vector $\mathbf{r}$ was evaluated for every individual in the OASIS-3 dataset. Also, the population of residual vectors for the HC group is denoted by $\mathbf{r}_{HC}$, and that for individuals in the AD+ group by $\mathbf{r}_{AD+}$. The length of the residual vectors is the same as the number of cortical thickness features (i.e., $m = 148$). Further, each element of the residual vector was mapped to a distinct brain region and ANOVA was used to test for group differences between individuals in HC and AD+ groups. Also, since elevation in $\Delta$-Age is the biomarker of interest in this analysis, we hypothesized that the brain regions that exhibited higher means for regional residuals for AD+ group than HC group would be the most relevant to capturing accelerated aging. Hence, the results are reported only for brain regions that showed elevated regional residual distribution in AD+ group with respect to HC group. Further, the group difference between AD+ and HC groups in the residual vector element for a brain region was deemed significant if it met the following criteria: i) the corrected $p$-value (Bonferroni correction) for the clinical diagnosis label in the ANOVA model was smaller than $0.05$; and ii) the uncorrected $p$-value for clinical diagnosis label in ANCOVA model with age and sex as covariates was smaller than $0.05$. See Appendix H for an example of this analysis.

Recall that 100 distinct VNNs were trained as regression models on different permutations of the training set of cortical thickness features from HC group. We used these trained models to establish the robustness of observed group differences in the distributions of regional residuals.

**Deriving anatomical interpretability or regional profile for $\Delta$-Age from the robustness of findings from regional analyses.** We performed the regional analysis described above corresponding to each trained VNN model and tabulated the number of VNN models for which a brain region was deemed to be associated with a significantly elevated regional residual for the AD+ group. A larger number of VNN models isolating a brain region as significant suggested that this region was likely to be a highly robust contributor to accelerated aging in the AD+ group.

**Explaining the anatomical interpretability.** We further investigated the relationship between regional residuals derived from VNNs and the eigenvectors of $\mathbf{C}_{HA}$ to determine the specific eigenvectors (principal components) of $\mathbf{C}_{HA}$ that were instrumental to anatomical interpretability. For this

purpose, we evaluated the inner product of normalized residual vectors (norm = 1) obtained from VNNs and the eigenvectors of the covariance matrix $\mathbf{C}_{\mathsf{HA}}$ for the individuals in AD+ group. The normalized residual vector is denoted by $\bar{\mathbf{r}}_{\mathsf{AD+}}$. For every individual, the mean of the absolute value of the inner product $|<\bar{\mathbf{r}}_{\mathsf{AD+}}, \mathbf{v}_i>|$ (where $\mathbf{v}_i$ is the $i$-th eigenvector of $\mathbf{C}_{\mathsf{HA}}$) was evaluated for the 100 VNN models.

Note that the VNNs were trained as described in Section 3.1 and hence, their ability to exploit the eigenvectors of the covariance matrix in a specific manner was learned when trained to predict chronological age for a healthy population. Hence, to gauge whether the anatomical interpretability was contingent on the learned ability of VNNs to exploit the eigenvectors of the covariance matrix from the procedure in Section 3.1, we also derived the anatomical interpretability for randomly initialized VNNs.

### 3.3 Individual-level brain age prediction

Finally, a scalar estimate for the brain age was obtained from the VNN regression output through a procedure consistent with the existing studies in this domain. Note that 100 VNNs provide 100 estimates $\hat{y}$ of the chronological age for each subject. For simplicity, we consider $\hat{y}$ to be the mean of these estimates. A systemic bias in the gap between $\hat{y}$ and $y$ may potentially exist when the correlation between $\hat{y}$ and $y$ is smaller than 1. Such a bias can confound the interpretations of brain age due to underestimation for older individuals and overestimation for younger individuals [45]. Therefore, to correct for this age-induced bias in $\hat{y} - y$, we adopted a linear regression model-based approach [45, 46]. Specifically, the following bias correction steps were applied on the VNN estimated age $\hat{y}$ to obtain the brain age $\hat{y}_{\mathsf{B}}$ for an individual with chronological age $y$:

**Step 1.** Fit a regression model for the HC group to determine scalars $\alpha$ and $\beta$ in the following model:
$$\hat{y} - y = \alpha y + \beta . \tag{8}$$

**Step 2.** Evaluate brain age $\hat{y}_{\mathsf{B}}$ as follows:
$$\hat{y}_{\mathsf{B}} = \hat{y} - (\alpha y + \beta) . \tag{9}$$

The gap between $\hat{y}_{\mathsf{B}}$ and $y$ is the $\Delta$-Age and is defined below. For an individual with cortical thickness $\mathbf{x}$ and chronological age $y$, the brain age gap $\Delta$-Age is formally defined as
$$\Delta\text{-Age} \triangleq \hat{y}_{\mathsf{B}} - y , \tag{10}$$

where $\hat{y}_{\mathsf{B}}$ is determined from the VNN age estimate $\hat{y}$ and $y$ according to steps in (8) and (9). The age-bias correction in (8) and (9) was performed only for the HC group to account for bias in the VNN estimates due to healthy aging, and then applied to the AD+ group. Further, the distributions of $\Delta$-Age were obtained for all individuals in HC and AD+ groups.

$\Delta$-Age for AD+ group was expected to be elevated as compared to HC group as a consequence of elevated regional residuals derived from the VNN model. To elucidate this, consider a toy example with two individuals of the same chronological age $y$, with one belonging to the AD+ group and another to the HC group. Equation (9) suggests that their corresponding VNN outputs (denoted by $\hat{y}_{\mathsf{AD+}}$ for individual in the AD+ group and $\hat{y}_{\mathsf{HC}}$ for individual in the HC group) are corrected for age-bias using the same term $\alpha y + \beta$. Hence, $\Delta$-Age for the individual in the AD+ group will be elevated with respect to that from the HC group only if the VNN prediction $\hat{y}_{\mathsf{AD+}}$ is elevated with respect to $\hat{y}_{\mathsf{HC}}$. Since the VNN predictions $\hat{y}_{\mathsf{AD+}}$ and $\hat{y}_{\mathsf{HC}}$ are proportional to the unweighted aggregations of the estimates at the regional level [see (4) and (5)], larger $\hat{y}_{\mathsf{AD+}}$ with respect to $\hat{y}_{\mathsf{HC}}$ can be a direct consequence of a subset of regional residuals [see (7)] being robustly elevated in AD+ group with respect to HC group across the 100 VNN models. When the individuals in this example have different chronological age, the age-bias correction is expected to remove any variance due to chronological age in $\Delta$-Age. We also verified that the differences in $\Delta$-Age for AD+ and HC group were not driven by age or sex differences via ANCOVA with age and sex as covariates.

## 4 Results

### 4.1 Chronological age prediction for the HC group

The performance achieved by the VNNs on the chronological age prediction task for the HC group has been summarized over the 100 nominal VNN models. VNNs achieved a mean absolute error (MAE)

of $5.82 \pm 0.13$ years and Pearson's correlation of $0.51 \pm 0.078$ between the chronological age estimates and the ground truth on the test set. Moreover, on the complete dataset, the MAE was $5.44 \pm 0.18$ years and Pearson's correlation was $0.47 \pm 0.074$. Thus, the trained VNNs were not overfit on the training set.

Next, for every individual in the HC group, we evaluated the mean of the inner products (also equivalently referred to as dot product) between the vectors of contributions of every brain region [$\mathbf{p}$ in (5)] and the eigenvectors of the anatomical covariance matrix for all 100 VNN models. The strongest alignment was present between the first eigenvector of the anatomical covariance matrix (i.e., the eigenvector associated with the largest eigenvalue) and the vectors of regional contributions to the VNN output ($0.987 \pm 0.0005$ across the HC group), with relatively smaller associations for second ($0.051 \pm 0.003$), third ($0.075 \pm 0.004$), and fourth ($0.094 \pm 0.003$) eigenvectors. Additional details are included in Appendix G. Thus, the VNNs exploited the eigenvectors of the anatomical covariance matrix to achieve the learning performance in this task. The first eigenvector of the anatomical covariance matrix predominantly included bilateral anatomic brain regions in the parahippocampal gyrus, precuneus, inferior medial temporal gyrus, and precentral gyrus.

We remark that several existing studies on brain age prediction have utilized deep learning and other approaches to report better MAE on their respective healthy populations [20–22, 47]. In contrast, our contribution here is conceptual, where we have explored the properties of VNNs when they are trained to predict the chronological age for the HC group. Subsequently, our primary focus in the context of brain age is on demonstrating the anatomical interpretability offered by VNNs and relevance of eigenvectors of the anatomical covariance matrix. Thus, we further provide the insights that have not been explored (or are infeasible to obtain) for most existing brain age evaluation frameworks based on less transparent deep learning models.

## 4.2 Analyses of regional residuals derived from VNNs revealed regions characteristic of AD

Figure 2a displays the robustness (determined via analyses of 100 VNN models) for various brain regions being associated with significantly larger residual elements for the AD+ group than the HC group. The most significant regions with elevated regional residuals in AD+ with respect to HC were concentrated in bilateral inferior parietal, temporal, entorhinal, parahippocampal, subcallosal, and precentral regions. All these brain regions, except for precentral and subcallosal, mirrored the cortical atrophy (Fig. 11 in supplementary material), and these regions are known to be highly interconnected with hippocampus [48]. Hence, brain regions characteristic of AD had significant differences in regional residual distributions for AD+ group as compared to HC group.

Although the results in Fig. 2a provided a meaningful regional profile for AD+ group, we further performed exploratory analysis to check whether the regional residuals had any clinical significance. To this end, we evaluated the Pearson's correlations between CDR sum of boxes and the regional residuals derived from final layer VNN outputs for the AD+ group for all 100 VNN models. Interestingly, the brain regions with the largest correlations with the CDR sum of boxes scores in Fig. 2c were concentrated in the parahippocampal, medial temporal lobe, and temporal pole regions (also isolated in Fig. 2a). This observation provides the evidence that the regional residuals for the AD+ group that led to the result in Fig. 2a could predict dementia severity.

## 4.3 $\Delta$-Age was elevated in AD+ group and correlated with CDR

We evaluated the $\Delta$-Age for HC and AD+ groups according to the procedure specified in Section 3.3. We also investigated the Pearson's correlation between $\Delta$-Age and CDR sum of boxes scores in AD+ group. Figure 2b illustrates the distributions for $\Delta$-Age for HC and AD+ groups ($\Delta$-Age for HC: $0 \pm 2.83$ years, $\Delta$-Age for AD+: $3.54 \pm 4.49$ years). The difference in $\Delta$-Age for AD+ and HC groups was significant (Cohen's $d = 0.942$, ANCOVA with age and sex as covariates: $p$-value $< 10^{-20}$, partial $\eta^2 = 0.158$). Also, age and sex were not significant in ANCOVA ($p$-value $> 0.4$ for both). Hence, the group difference in $\Delta$-Age for the two groups was not driven by the difference in the distributions of their chronological age. Figure 2d plots CDR sum of boxes scores versus $\Delta$-Age for the AD+ group. Pearson's correlation between CDR sum of boxes score and $\Delta$-Age was $0.474$ ($p$-value $= 2.88 \times 10^{-12}$), thus, implying that the $\Delta$-Age evaluated for AD+ group captured information about dementia severity. Hence, as expected, the $\Delta$-Age for AD+ was likely to be larger

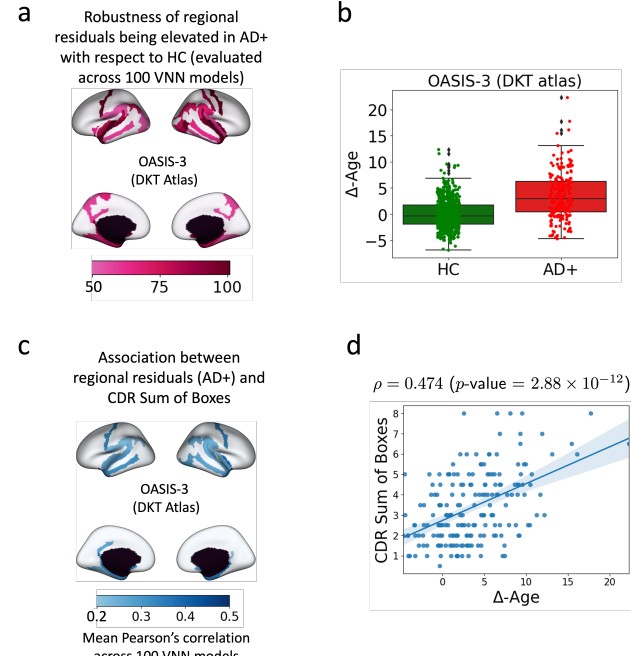

Figure 2: **Anatomically interpretable Δ-Age evaluation in OASIS-3.** Panel **a** displays the robustness of the significantly elevated regional residuals for AD+ group with respect to HC group for different brain regions. For every VNN model in the set of 100 nominal VNN models that were trained on HC group, the analyses of regional residuals helped isolate brain regions that corresponded to significantly elevated regional residuals in AD+ group with respect to HC group. After performing this experiment for 100 VNN models, the robustness of the observed significant effects in a brain region was evaluated by calculating the number of times a brain region was identified to have significantly elevated regional residuals in AD+ group with respect to HC group. The number of times a brain region was linked with significantly elevated regional residuals in AD+ group with respect to HC group is projected on the brain template. Panel **b** displays the distribution of Δ-Age for HC and AD+ groups. The elevated brain age effect here is characterized by regional profile in Panel **a**. Panel **c** projects the mean Pearson's correlation between regional residuals (derived for each VNN model in the set of 100 nominal VNN models) and CDR sum of boxes for AD+ group on the brain template. Panel **d** displays the scatter plot for CDR sum of boxes versus Δ-Age in AD+ group. The correlation between Δ-Age and CDR sum of boxes could be attributed to the observations in Panel **c**.

with an increase in CDR sum of boxes scores. For instance, the mean Δ-Age for individuals with CDR sum of boxes greater than 4 was $6.04$ years, and for CDR sum of boxes $\leq 4$ was $2.42$ years.

Given that the age-bias correction procedure is a linear transformation of VNN outputs, it can readily be concluded that the statistical patterns for regional residuals in Fig. 2a and Fig. 2c lead to elevated Δ-Age and correlation between Δ-Age and CDR sum of boxes scores. Therefore, our framework provides a feasible way to characterize accelerated biological aging in AD+ group with a regional profile. Additional figures and details pertaining to VNN outputs and brain age before and after the age-bias correction was applied are included in Appendix J.

### 4.4 Regional residuals derived from VNNs were correlated with eigenvectors of the anatomical covariance matrix

Figure 3a plots the mean inner product $|<\bar{\mathbf{r}}_{AD+}, \mathbf{v}_i>|$ for eigenvectors associated with 50 largest eigenvalues of $\mathbf{C}_{HA}$. The three largest mean correlations with the regional residuals in AD+ group were observed for the third eigenvector of $\mathbf{C}_{HA}$ ($|<\bar{\mathbf{r}}_{AD+}, \mathbf{v}_3>| = 0.645 \pm 0.016$), fourth eigenvector ($|<\bar{\mathbf{r}}_{AD+}, \mathbf{v}_4>| = 0.305 \pm 0.02$), and the first eigenvector ($|<\bar{\mathbf{r}}_{AD+}, \mathbf{v}_1>| = 0.299 \pm 0.001$). These eigenvectors are plotted on a brain template in the expanded Fig. 3b. Inspection of the first,

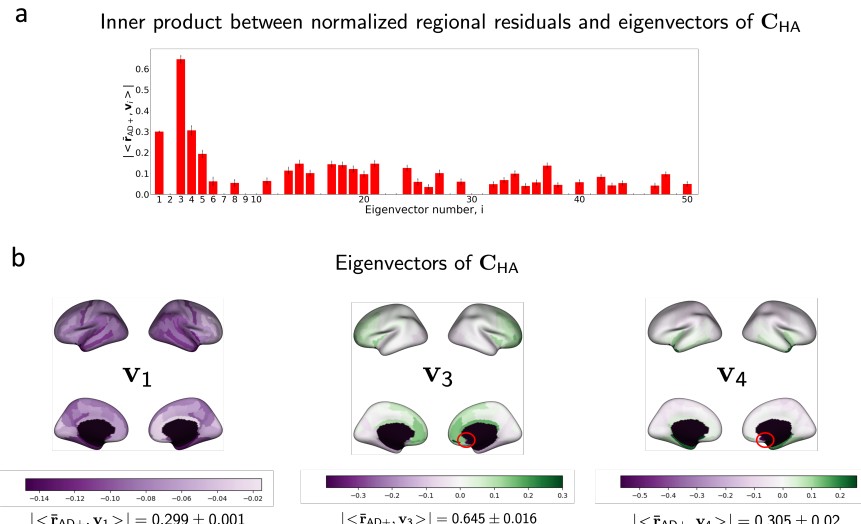

Figure 3: **a.** Bar plots of the mean inner products between the normalized vector of regional residuals (norm = 1) of VNN outputs (VNNs trained on OASIS-3) obtained from AD+ group and the eigenvectors of $\mathbf{C}_{HA}$ (covariance matrix of combined HC and AD+ group) with respective standard deviations as error bars. Results with coefficient of variation $> 30\%$ across the AD+ group have been excluded. **b.** The eigenvectors associated with the top three largest values for $|<\bar{\mathbf{r}}_{AD+}, \mathbf{v}_i>|$ are plotted on the brain surface. Subcallosal region in the right hemisphere was associated with the element with the largest magnitude in $\mathbf{v}_3$ and $\mathbf{v}_4$ and is highlighted with a red circle in the corresponding plots.

third, and fourth eigenvectors of $\mathbf{C}_{HA}$ suggested that subcallosal, entorhinal, parahippocampal and temporal pole regions had the most dominant weights (in terms of magnitude) in these eigenvectors.

### 4.5 Anatomical interpretability was diminished for randomly initialized VNNs.

Finally, we leveraged 100 VNNs that were randomly initialized (i.e., not trained whatsoever) to evaluate the regional profiles for brain regions that exhibited elevated regional residuals for AD+ group with respect to HC group in OASIS-3. These VNNs had the same architecture as the VNNs that were trained on OASIS-3. Figure 13a in the supplementary material shows that the robustness of the regional residuals being elevated for AD+ group with respect to HC group was severely diminished as compared to the parallel results in Fig. 2a. Similarly, the correlation between the regional residual derived using randomly initialized VNNs and CDR scores was highly inconsistent (Fig. 13b in supplementary material) as compared to parallel results in Fig. 2c. These findings suggested that the ability of VNNs to exploit the covariance matrix (learned when trained to predict chronological age of the HC group) was instrumental to the results derived in Fig. 2.

### 4.6 Additional Experiments

**Cross-validation:** The results derived in Fig. 2a and Fig. 2b were cross-validated on the ADNI-1 dataset (see Appendix I for details).

**Stability to perturbations in $\mathbf{C}_{HA}$:** As a consequence of the stability of VNNs, we observed that the regional profile for $\Delta$-Age in Fig. 2a was stable even when the covariance matrix $\mathbf{C}_{HA}$ was estimated by a variable composition of individuals from the HC and AD+ group (Appendix M).

**Anatomical covariance matrix and brain age:** Use of anatomical covariance matrix derived from only HC group provides results consistent with Fig. 2, albeit with a slightly smaller group difference between the $\Delta$-Age distributions for HC and AD+ groups. See Appendix N for details.

We refer the reader to Appendix C for discussions on various aspects of this work and its limitations and associated societal concerns.

# 5  Acknowledgements

OASIS-3 data were provided by Longitudinal Multimodal Neuroimaging: Principal Investigators: T. Benzinger, D. Marcus, J. Morris; NIH P30 AG066444, P50 AG00561, P30 NS09857781, P01 AG026276, P01 AG003991, R01 AG043434, UL1 TR000448, R01 EB009352. Data collection and sharing for ADNI-1 dataset was funded by the Alzheimer's Disease Neuroimaging Initiative (ADNI) (National Institutes of Health Grant U01 AG024904) and DOD ADNI (Department of Defense award number W81XWH-12-2-0012). ADNI data are disseminated by the Laboratory for Neuro Imaging at the University of Southern California.

We are grateful to the anonymous reviewers, whose suggestions have greatly improved this paper.

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

# A    Data and Code Availability

OASIS-3 dataset is publicly available and hosted on `central.xnat.org`. Access to OASIS-3 dataset may be requested through `https://www.oasis-brains.org/`. Supplementary files include (i) the VNN models trained to predict chronological age for HC group, (ii) code for demonstrations of evaluating regional profiles corresponding to elevated regional residuals in AD+ group, and correlations between the eigenvectors of the anatomical covariance and VNN outputs using a small subset of $z$-score normalized cortical thickness data, (iii) the regional residuals derived from different VNN models for the OASIS-3 dataset that lead to results in Fig. 2, and (iv) the code for brain age evaluation on OASIS-3 dataset. All material is also available online at https://github.com/sihags/VNN_Brain_Age.

# B    Relevant Literature

**Graph convolutional networks.** GCNs typically rely on an information aggregation procedure (referred to as graph convolutions) over a graph structure for data processing. Several implementation strategies for graph convolution operations have been proposed in the literature, including spectral convolutions [49], Chebyshev polynomials [50], ordinary polynomials [36], and diffusion based representations [51]. GCNs admit the properties of stability to topological perturbations and transferability across graphs of different sizes in various settings [38, 52–54], which makes them a well-motivated data analysis tool for graph-structured data.

In [31], coVariance neural networks (VNN) were proposed as GCNs with sample covariance matrices as graph and polynomial graph filters as convolution operation. Covariance matrices and principal component analysis (PCA) form the two cornerstones of non-parametric analyses in real world applications that have spatially distributed, multi-variate data acquisition protocols, including neuroimaging [19], computer vision [55, 56], weather modeling [57], traffic flow analysis [58], and cloud computing [59].

**Explainability in GNNs.** We refer the reader to [32, 60] for a detailed review on explainability in GNNs. Here, we adopt the taxonomy from [32], which categorizes the recent efforts to add explainability to GNN models into two categories: instance-level methods and model-level methods for explainability. Instance-level methods for explainability are extensions of the standard model-agnostic methods for wider categories of deep learning models to GNNs, and aim to identify features most important to the inference outcome. Examples of techniques used to determine the importance of features include gradient-based methods [61], perturbation-based methods [62], and surrogate methods [63]. Such methods are, in principle, input-dependent as they provide explanations based on the given instance of the dataset. The robustness of the conclusions drawn from such methods to perturbations in the input and variations in the model training algorithms is an active area of research.

Model-level explainability has previously been studied in terms of graph topology in [64]. Further, importance of subgraphs to inference outcomes using Shapley values was studied in [65]. Since graph topology informs convolution operations in GNNs, it can provide a more generic explanation to GNNs than those that focus on individual features (such as nodes or edges of the graph). Since the convolution operation in VNNs is equivalent to manipulating the input data according to the eigenvectors of the covariance matrix [31], the eigenvectors of the covariance matrix are instrumental to explaining the inference by VNNs. Hence, we argue that VNNs can offer model-level explainability, similar in spirit as discussed in [32], where the explainability hinges on the eigenvectors of the covariance matrix.

**Interpretable brain age prediction.** Limited focus has been on comparable studies that associate brain age gaps with regional profiles [20, 66]. The study in [20] adopts a convolutional neural network approach to infer brain age from MRI images directly and assigns importance to brain regions in evaluating the brain age. In principle, the interpretability offered by VNNs in the context of brain age is similar, as we infer a regional profile for $\Delta$-Age by isolating the brain regions that are contributors to the elevated $\Delta$-Age in neurodegeneration. In addition, the regional profile identified by VNNs is correlated with specific eigenvectors or the principal components of the anatomical covariance matrix. Hence, the $\Delta$-Age inferred by our framework is driven by the ability of a VNN to manipulate the input data according to certain principal components of the anatomical covariance matrix. Also, VNNs are significantly less complex deep learning models as compared to those studied in [20]. Our results demonstrate that the VNNs trained with less than 300 learnable parameters

exhibit regional interpretability in the context of brain age in AD. In general, the regional expressivity offered by VNNs is in stark contrast to a multitude of existing relevant studies that rely on less transparent statistical approaches and further use post-hoc analyses (such as ablation analysis [67–69] or exploring correlations with region-specific markers [25] and psychiatric symptoms [47, 70]) to assign interpretability to a scalar, elevated $\Delta$-Age effect.

## C  Discussion

Our study has proposed a methodologically transparent framework for brain age prediction using VNNs. In contrast to existing studies that primarily focus on the performance of the model in predicting the chronological age of a healthy population, we have focused on the properties that VNNs gained when trained for this task. In particular, the VNNs achieve learning by transforming the input data according to the principal components or the eigenvectors of the covariance matrix estimated from the data. Hence, training the VNNs to predict chronological age using cortical thickness features from a healthy population had enabled them to exploit the eigenvectors of the anatomical covariance matrix in a specific manner. Further, the anatomical interpretability associated with elevated $\Delta$-Age in Alzheimer's disease was derived by the statistical analyses of the features extracted at the final layer of the VNNs and projecting the results of these analyses on the brain surface. Thus, the anatomical interpretability offered by VNNs is an inherent feature of VNNs and fundamentally distinct from the state-of-the-art model agnostic explainability approaches in the context of brain age (such as SHAP, LIME, saliency maps etc., that provide importance weights to individual input features).

The lack of explainability in the machine learning models deployed in brain age prediction frameworks may be a fundamental reason behind exclusive focus on the performance-driven approaches in this domain thus far. VNNs are inherently explainable models, as their inference outcomes can be tied with the eigenvectors of the covariance matrix. Hence, the anatomical interpretability offered by VNNs to $\Delta$-Age could be explained by their ability to exploit specific eigenvectors of the anatomical covariance matrix. This observation is highly relevant, as the quality of prediction on the chronological prediction for healthy population by itself may not be a complete determinant of the quality of brain age prediction in neurodegeneration. Furthermore, the role of the age-bias correction step in the VNN-based brain age prediction framework was restricted to projecting the VNN outputs onto a space where one could infer accelerated biological aging with respect to the chronological age from a layman's perspective.

By associating $\Delta$-Age with a regional profile, VNNs also provide a feasible tool to distinguish pathologies if the distributions of $\Delta$-Age for them are overlapping. A larger focus is needed on principled statistical approaches for brain age prediction that can capture the factors that lead to accelerated aging. Locally interpretable and theoretically grounded deep learning models such as VNNs can provide a feasible, promising future direction to build statistically and conceptually legitimate brain age prediction models in broader contexts. Incorporating other modalities of neuroimaging or alternative metrics of aging other than chronological age (such as DNA methylation [71]) provide promising future directions that can help improve our understanding of aging.

**Limitations.** Existing studies, including this paper, fall short at concretely defining the notion of optimal brain age. From a broader perspective, quantifying biological age even for a healthy population is a complex task due to various factors that can contribute to accelerated aging in the absence of an adverse health condition [72–74]. Moreover, the impacts of the quality of MRI scans and brain atlases across datasets on $\Delta$-Age must also be explored.

Our analysis was limited to older individuals, and a dataset with more diverse age groups is expected to provide holistic information on brain age. Isolation of brain regions contributing more to $\Delta$-Age in AD than HC hinges on a binary group comparison. Such a comparison can be impacted by the composition of the dataset (for instance, a skewed dataset may not provide informative results).

**Societal Concerns.** We are grateful to the reviewers for suggesting the potential societal concerns that may arise due to brain age prediction algorithms in practice. These societal concerns are discussed next. While the adoption of brain age prediction algorithms in clinic can lead to better decision-making, it is also worth considering that inaccurate prediction of a brain age gap (given implications for neurodegenerative diseases) could potentially lead to unintended consequences (such as over-medication in the case where the brain age gap is over-estimated, as just one example). Furthermore,

any framework for predicting brain age in an anatomically interpretable manner could be abused in potentially unethical or dubious ways for commercial or sociopolitical reasons. For instance, using such predictions to curb immigration and asylum or increasing health insurance premiums can be two such examples.

## D  An Abstract Overview of VNN-based Brain Age Prediction

Figure 4 provides an abstract overview of the general procedure of evaluating brain age using machine learning (ML) models. From Fig. 4, we note that if the ML model is a black box, it may be infeasible to capture the contributors to elevated age-gap in Step 3. Furthermore, in this context, it is also unclear whether age-bias correction step (Step 2) influences final $\Delta$-Age prediction through some statistical artifact [23]. Hence, it can be desirable to minimize the role of age-bias correction in $\Delta$-Age evaluation by selecting an ML model that achieves a near perfect fit on chronological age of healthy controls in Step 1. However, there is no guarantee that achieving an 'perfect fit' on true age of healthy controls will enable the ML model to capture the impact of neurodegeneration in individuals with neurodegeneration.

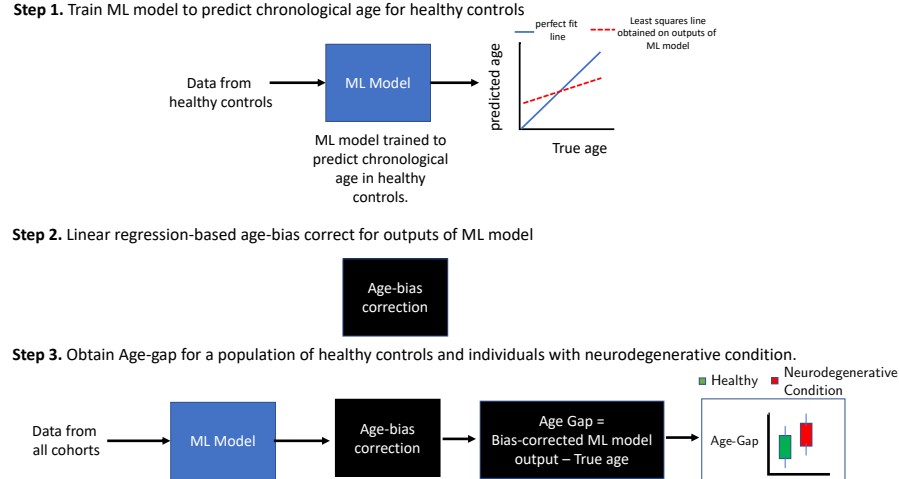

Figure 4: **A general overview of brain age evaluation using machine learning algorithms in the existing literature. Step 1** consists of training a machine learning (ML) model to predict chronological age (true age) for healthy controls. If the correlation between predicted age and true age is smaller than 1, an age-bias exists in ML model outputs as the age for older individuals tends to be under-estimated and that for younger individuals tends to be over-estimated. To correct for this bias, a linear regression based model is applied on the ML model outputs in **Step 2**. Under the hypothesis that ML model can capture accelerated aging in neurodegeneration, it is expected that $\Delta$-Age for individuals with neurodegeneration will be significantly higher than those of healthy controls (**Step 3**).

VNNs allow us to analyze the contribution of each feature (brain region) to the final output. Hence, by analyzing the elevations in contributions of different brain regions via studying group differences in regional residuals, we are able to characterize the brain regions that contribute to accelerated aging (or larger $\Delta$-Age) in individuals with neurodegeneration (Fig. 5). Thus, we can verify that VNNs captured neurodegeneration-driven effects that eventually led to elevated $\Delta$-Age for an individual. Our experiments show that VNNs do not obtain a perfect fit on chronological age of healthy individuals. Hence, age-bias correction is important to appropriately project the VNN model outputs via a linear model into an appropriate space such that a clinician can observe an elevated $\Delta$-Age effect in individuals with neurodegenerative condition (AD in this paper). Based on these observations, we remark that VNNs provide an interpretable framework for brain age prediction.

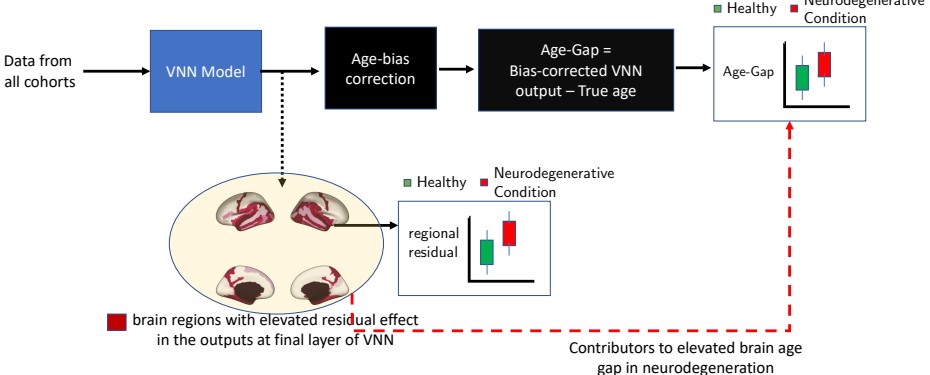

Figure 5: **Interpretability offered by VNNs in brain age prediction.** By analyzing the final layer outputs of VNNs, we can isolate brain regions that have larger regional residuals for individuals with AD with respect to healthy controls. Furthermore, the elevated regional residuals in these brain regions eventually contribute to elevated $\Delta$-Age after age-bias correction.

# E  VNN Architecture

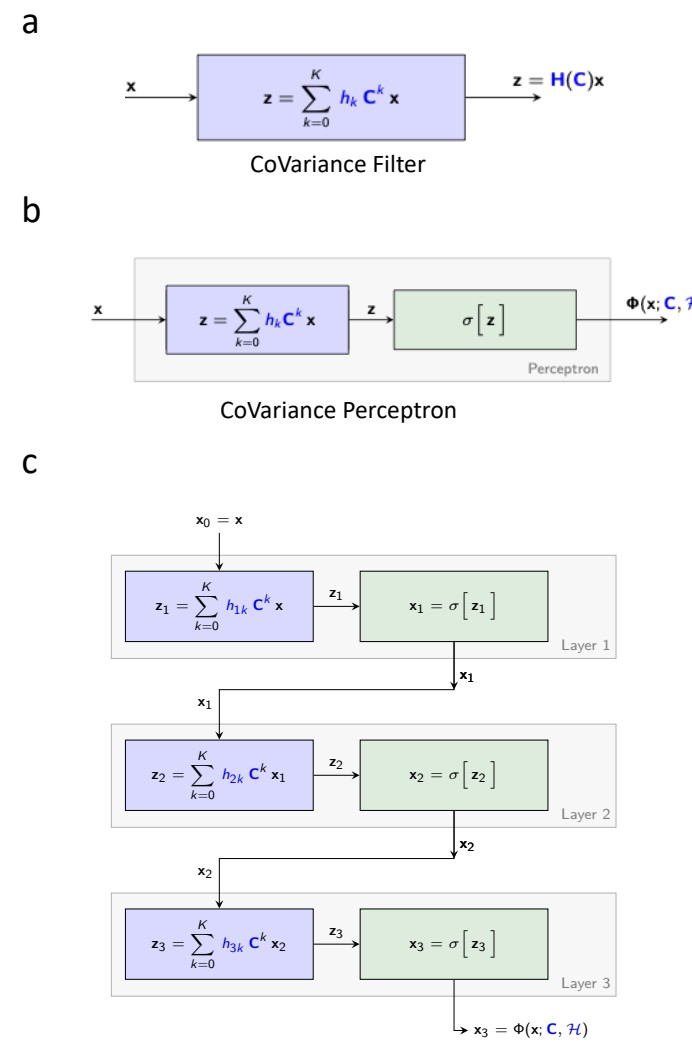

Figure 6: **Basics of VNN architecture.** Panel **a** illustrates that the coVariance filter $\mathbf{H(C)}$ is a polynomial in $\mathbf{C}$ and its application on input $\mathbf{x}$. Panel **b** shows the construction of a coVariance perceptron based on coVariance filter $\mathbf{H(C)}$ and pointwise nonlinearity $\sigma$. coVariance perceptron specifies one layer of VNN. Panel **c** shows a basic multi-layer VNN architecture formed by stacking three coVariance perceptrons.

## F    VNN training

We randomly split the HC group into an approximately $90/10$ train/test split. Thus, the test set consisted of $61$ healthy individuals. The sample covariance matrix was evaluated using all samples in the training set ($n = 550$). Furthermore, this covariance matrix was normalized such that its maximum eigenvalue was $1$. Cortical thickness data was $z$-score normalized across the training set and this normalization was applied to the test set. Next, the training set was randomly split internally, such that, the VNN was trained with respect to the mean squared error loss between the predicted age and the true age in $n = 489$ samples of the HC group. The loss was optimized using batch stochastic gradient descent with Adam optimizer available in PyTorch library [75] for up to $100$ epochs. The batch size was $78$ (determined via 'optuna' package [44]). The VNN model with the best minimum mean squared error performance on the remaining $61$ samples in the training set (which acted as a validation set) was included in the set of nominal models for this permutation of the training set. For each dataset, we trained and validated the VNN models over $100$ permutations of the complete training set of $n = 550$ samples for the HC group, thus, leading to $100$ trained VNN models (also referred to as nominal models) per dataset.

## G    VNN regression model outputs for HC group in OASIS-3 are correlated with the first eigenvector of anatomical covariance matrix

The study in [31] suggests that VNN based statistical inference draws conceptual similarities with PCA-driven analysis. Hence, we further investigated whether the regression performance achieved by VNNs in predicting the chronological age of HC group could be characterized by contributions of the eigenvectors of the anatomical covariance matrix. To avoid any selection bias, we report the results on the complete HC group, where the cortical thickness features were $z$-score normalized across the group such that the mean of cortical thickness for a brain region across the HC group was $0$. Here, the notation $\mathbf{C_H}$ denotes the anatomical covariance matrix derived from the complete HC group.

Recall that the final regression output by VNNs is formed by an unweighted average function as a readout function. Thus, we can equivalently represent the functionality of the readout as a simple aggregation of the contributions of different features or brain regions to the final estimate formed by the VNN (see (5)). Hence, for every individual, we evaluated the mean of the inner products (also equivalently referred to as dot product between vectors) between the vectors of contributions of every brain region with the eigenvector of the covariance matrix $\mathbf{C_H}$ for all $100$ VNN models. Note that a vector of regional contributions was of the same length as the number of cortical thickness features (i.e., $148$ for OASIS-3) and therefore, each element of this vector was associated with a distinct brain region. We use the notation $\mathbf{p_{HC}}$ to represent the population of vectors obtained from the HC group. To evaluate the inner product, we used $\bar{\mathbf{p}}_{HC}$, which was obtained from $\mathbf{p_{HC}}$ after normalization (norm $= 1$). We denote the population of inner products across the HC group in OASIS-3 by $|<\bar{\mathbf{p}}_{HC}, \mathbf{v}_i >|$ for an eigenvector $\mathbf{v}_i$ of $\mathbf{C_H}$. Note that since all vectors in $\bar{\mathbf{p}}_{HC}$ were normalized to have norm $1$ and the eigenvectors of $\mathbf{C_H}$ were of length $1$ by default, $|<\bar{\mathbf{p}}_{HC}, \mathbf{v}_i >|$ represented the population of the cosine of the angles between the vectors in $\mathbf{p_{HC}}$ and eigenvectors $\mathbf{v}_i$ of $\mathbf{C_H}$ across the HC group in OASIS-3. Figure 7a plots the mean of the inner products observed across the HC group for the first $30$ eigenvectors of $\mathbf{C_H}$ for all $100$ VNNs. Figure 7b illustrates the projection of $\mathbf{v}_1$ on a brain template.

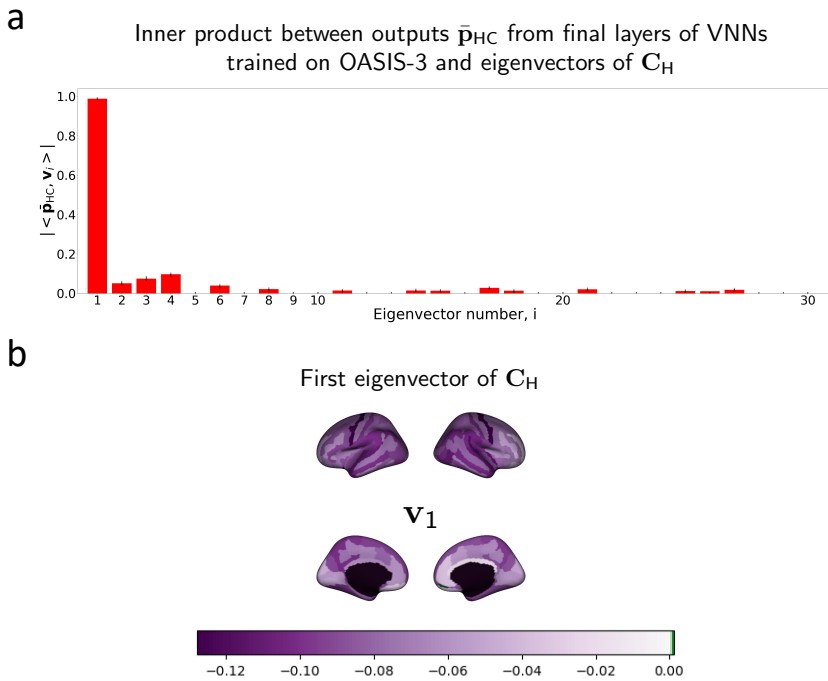

Figure 7: **Inner product between the normalized vector of regional contributions to the VNN outputs ($\bar{\mathbf{p}}_{HC}$) and eigenvectors of $\mathbf{C}_H$ (anatomical covariance matrix for HC group in OASIS-3).** Panel **a** illustrates a bar plot for $|<\bar{\mathbf{p}}_{HC}, \mathbf{v}_i>|$ for $i \in \{1, \ldots, 30\}$, where $\mathbf{v}_i$ is the $i$-th eigenvector (principal component) of covariance matrix $\mathbf{C}_H$ and associated with $i$-largest eigenvalue in terms of magnitude and the vectors of regional contributions, $\bar{\mathbf{p}}_{HC}$ were obtained by VNNs that were trained on OASIS-3 dataset. The inner product results for eigenvectors with coefficient of variation greater than 30% across the HC group of OASIS-3 were excluded (and hence, their respective entries set as 0). For every individual in HC group, the associations between their corresponding vector of regional contributions, $\bar{\mathbf{p}}_{HC}$ and eigenvectors of $\mathbf{C}_H$ were evaluated over 100 nominal VNN models. The first eigenvector ($\mathbf{v}_1$) had the largest association. The eigenvector $\mathbf{v}_1$ is plotted on a brain template in Panel **b**.

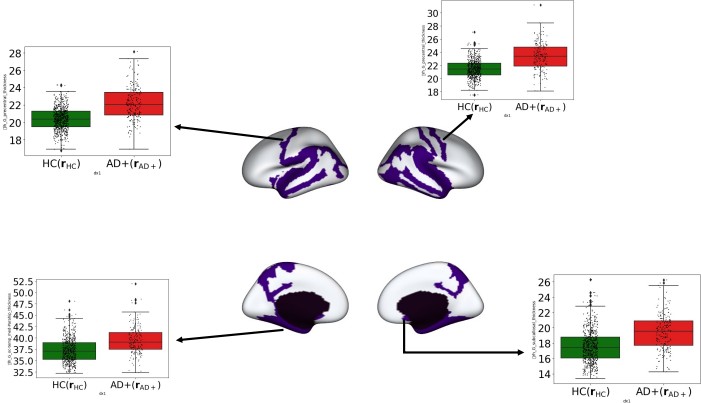

Figure 8: Results depicting the brain regions with significantly elevated regional residuals for AD+ group with respect to HC group in OASIS-3. The results here were derived by a VNN model that was trained as a regression model to predict chronological age from cortical thickness data for HC group in OASIS-3. Box plots depicting the distributions of regional residuals in the HC and AD+ groups are shown for a few representative regions.

# H    Illustration of regional residual analysis from VNN model outputs

In this section, we demonstrate the regional analysis described in Section 3.2 for a VNN model that was trained to predict chronological age for HC group in OASIS-3 dataset. All mathematical notations referred to in this section are borrowed from Section 3.2. Note that no further training was performed for this VNN model to evaluate brain age or regional residuals.

The covariance matrix in this VNN model was replaced with $\mathbf{C}_{HA}$ derived from the cortical thickness features from both HC and AD+ groups. Further, the cortical thickness features in the HC group were $z$-score normalized and this normalization was used to transform the cortical thickness features of the AD group.

The age prediction $\hat{y}_i$ and a vector of residuals $\mathbf{r}_i$ were obtained for an individual $i \in \{1, \ldots, 805\}$ in the dataset. The size of residual vector $\mathbf{r}_i$ was $148 \times 1$ and hence, each element of $\mathbf{r}_i$ corresponded to a distinct brain region as defined by the DKT brain atlas with 148 parcellations. By evaluating the vector of residuals $\mathbf{r}_i$ for every individual in the combined dataset, a population of residual vectors from HC group (referred to as $\mathbf{r}_{HC}$) and AD+ group (referred to as $\mathbf{r}_{AD+}$) was constructed. The elements of these residual vectors are referred to as regional residuals throughout the paper.

Each dimension of these residual vectors was investigated for group differences between HC and AD+ groups via ANOVA as described in Section 3.2. Thus, for every VNN model, we eventually performed $m = 148$ number of ANOVA tests and evaluated the brain regions for significance in group differences in their respective residuals. The significance of group differences between the distributions of regional residuals for HC and AD+ groups corresponding to a brain region was determined after correcting the $p$-values of ANOVA test for multiple comparisons via Bonferroni correction (Bonferroni corrected $p$-value $< 0.05$). The group differences were additionally investigated for significance at an uncorrected level using ANCOVA with age and sex as covariates.

Figure 8 illustrates the results obtained via ANOVA in this context. The brain regions deemed significant according to the criteria provided in Section 3.2 have been shaded. The box plots for various brain regions show that the regional residuals were significantly elevated in AD+ group as compared to HC. The regional residuals that lead to the results in Fig. 8 are provided in the supplementary material (model ID 50) as part of the regional residuals extracted from all 100 VNN models that were trained on HC group in OASIS-3.

We had 100 trained VNN models for the OASIS-3 dataset and performed similar analyses for each of them. Further, we counted the number of models for which the above described analysis yielded a brain region to be significant. A brain region with robust group difference in its regional residual distribution in HC vs AD+ was expected to be more frequently labeled as significant by the VNN models. The results of this robustness analyses on the OASIS-3 dataset are shown in Fig. 2a.

# I    Cross-validation on ADNI-1 dataset

In this section, we provide results on the standardized 3.0 T ADNI1 dataset (see [34] for details), consisting of 47 controls (age = $75.06 \pm 3.93$ years, 29 females), 71 individuals with mild cognitive impairment (age = $74.03 \pm 8.12$ years, 26 females), and 33 individuals with dementia (age = $75.08 \pm 8.07$ years, 22 females) from the from the wider ADNI database. We chose this standardized dataset in the spirit of reproducibility and to avoid selection bias. The individuals with mild cognitive impairment and dementia diagnosis were combined to form the AD+ cohort, equivalent to that of the AD+ cohort for OASIS-3.

Data used from ADNI database were obtained from the Alzheimer's Disease Neuroimaging Initiative (ADNI) database. The ADNI was launched in 2003 as a public-private partnership, led by Principal Investigator Michael W. Weiner, MD. The primary goal of ADNI has been to test whether serial magnetic resonance imaging (MRI), positron emission tomography (PET), other biological markers, and clinical and neuropsychological assessment can be combined to measure the progression of mild cognitive impairment (MCI) and early Alzheimer's disease (AD). For up-to-date information, see www.adni-info.org.

**Data processing.** The MRI images for the 3.0T standardized ADNI-1 dataset at the baseline visit were downloaded from https://adni.loni.usc.edu/. Cortical thickness features (curated according to DKT atlas) were derived using the open-access CAT12 pipeline [76] using their default options. We refer the reader to https://neuro-jena.github.io/cat12-help/ for detailed processing steps. All outputs were quality checked visually for errors in grey matter segmentation.

**Cross-validation.** The results in Fig. 9 were obtained from the models that were trained on OASIS-3 dataset. Here, the anatomical covariance matrix was estimated using cortical thickness measures from all individuals in ADNI1 dataset. The observations on ADNI1 were highly consistent with those made in the paper. Specifically, in ADNI1 dataset, brain age gap was significantly higher for individuals with dementia as compared to healthy controls, with MCI in between them. Also, the subcallosal, entorhinal, temporal pole, and superior temporal regions were cross-validated on this dataset as being contributors to elevated $\Delta$-Age in the combined cohort of dementia and MCI (equivalent to AD+ group from OASIS-3) with respect to the HC group.

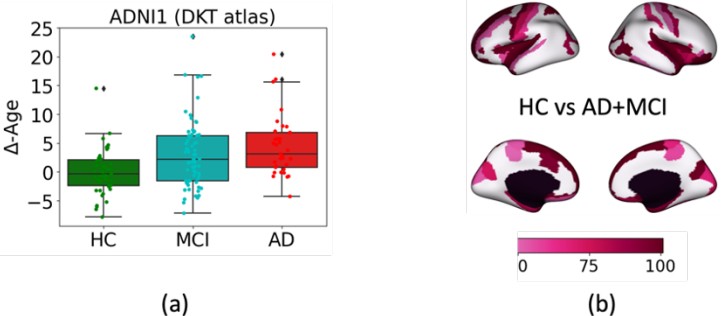

(a)                                    (b)

Figure 9: (a) Distribution of $\Delta$-Age across HC, MCI, Dementia cohorts derived from VNN models trained on OASIS-3. Anatomical covariance matrix from ADNI-1 dataset was used in the VNNs. $\Delta$-Age for controls: $0 \pm 3.92$ years, $\Delta$-Age for MCI $3 \pm 5.74$ years, $\Delta$-Age for AD: $4.49 \pm 5.34$ years. (b) Across 100 VNNs that had been trained on OASIS-3 dataset, we evaluated the number of times the regional residual mean was smaller for HC group than the AD+MCI group in ADNI-1 dataset.

## J  Additional details on brain age prediction in OASIS-3

In this section, we provide additional figures and discussions pertaining to the results for interpretable brain age prediction in Fig. 2. Figure 10 displays the distributions of chronological age for AD+ and HC groups in OASIS-3 dataset.

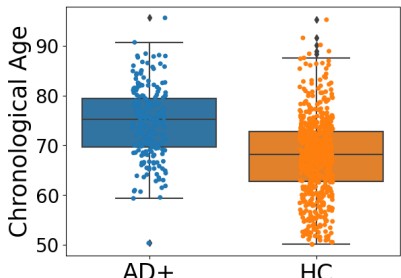

Figure 10: Distribution of chronological age in AD+ and HC groups.

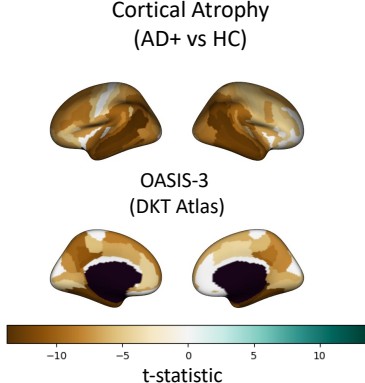

Figure 11: Results of group differences in cortical thickness between AD+ and HC groups. Regions with significant differences (two-sided t-test, Bonferroni corrected $p$-value $< 0.05$) are identified and the corresponding $t$-statistics are projected on a brain template. Negative $t$-statistic for a brain region suggests that the AD+ group had significant cortical atrophy in that region as compared to HC group.

Since VNNs were trained for the regression task, a VNN processed cortical thickness data and provided an estimate for the chronological age for each individual. Since we trained 100 VNN models on different permutations of the training set in OASIS-3, we use the mean of all VNN estimates as the VNN prediction for an individual. This VNN prediction is further leveraged to form brain age estimates and $\Delta$-Age metrics. Figure 12a displays the plot for VNN predictions versus chronological age (ground truth) for the complete HC group. The Pearson's correlation between VNN prediction and chronological age (ground truth) for HC group was $0.486$, which was similar to that reported in Section 4.1. VNN outputs clearly under-estimated the chronological age for older individuals and over-estimated the chronological age for individuals on the younger end of the age distribution for HC group.

Figure 12b displays the plot for VNN predictions versus chronological age (ground truth) for the complete AD+ group. The Pearson's correlation between VNN prediction and chronological age (ground truth) for AD+ group was $0.28$. We further note that the VNN architecture and our analysis of regional residuals helped quantify the contribution of each brain region to a data point in Fig. 12a and Fig. 12b. Hence, the scatter plot in Fig. 12b could be affected by larger contributions of certain brain regions for AD+ group relative to the HC group.

Figure 12c illustrates the box plots of residuals evaluated by the difference between VNN predictions and chronological age for HC and AD+ groups. Figure 12c suggests that the chronological age

for AD+ group was underestimated as compared to that for HC group. This observation was also expected since AD+ group is significantly older than the HC group. However, we expect that the robust elevated regional residuals from brain regions in Fig. 2b mitigated the under-estimation effect due to higher age of AD+ group to some extent.

Figures 12d-f display the results after age-bias correction is applied to the VNN outputs. As expected, the brain age for HC group in Fig. 12d is largely concentrated around the line of equality ($x = y$ line). In contrast, the brain age for AD+ group in Fig. 12e is concentrated above the line of equality. These effects manifest into the box plots for $\Delta$-Age in Fig. 12f where we observe the AD+ group to have elevated $\Delta$-Age as compared to HC group.

VNN architecture facilitated isolation of the effects of accelerated aging before age-bias correction was applied. Hence, the transformation of VNN outputs to brain age from Fig. 12a-c to Fig. 12d-f was not surprising. However, such insights may be infeasible for machine learning approaches that lack transparency and hence, the impact of deviations due to neurodegeneration from the healthy control population cannot be interpreted or isolated. In this context, if the learning model was a black box, Fig. 12a-c may appear to be counter-intuitive to the goal of detecting accelerated aging in the AD+ group and the effect of age-bias correction can be unclear, thus, leading to several criticisms [23].

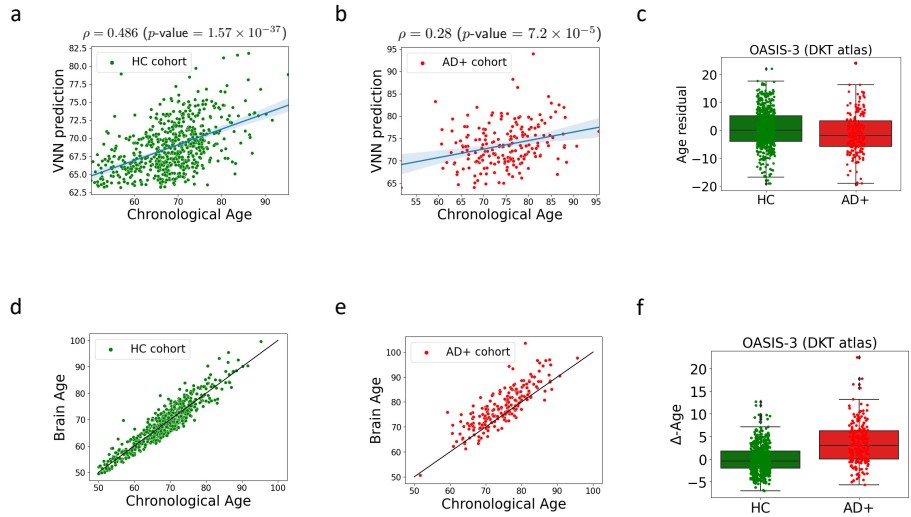

Figure 12: **Supplementary figures to Fig. 2.** Panel **a** displays the plot of VNN prediction versus chronological age for HC group. VNN predictions were obtained as the average of the outputs of 100 nominal VNNs that were trained on OASIS-3 and operated on the anatomical covariance matrix $\mathbf{C}_{\mathrm{HA}}$. Panel **b** displays the results similar to that in panel **a** for the AD+ group. The solid line in panels **a** and **b** is the least squares line. Panel **c** includes the boxplots for residuals derived from the difference between VNN predictions and chronological age for HC and AD+ groups. Panel **d** and **e** display the plots for brain age versus chronological age for HC and AD+ groups, respectively. The solid line in panels **d** and **e** is the identity line. Panel **f** displays the box plots for $\Delta$-Age in HC and AD+ groups.

## K   Adaptive readouts may penalize the interpretability of regional residuals and $\Delta$-Age

Thus far, we have focused on VNNs that operate with a non-adaptive readout (unweighted average) function. However, it is expected that the performance of the VNNs on their original task of chronological age prediction could be improved significantly with the help of an adaptive readout function. Our experiments showed that this was indeed the case. If a single-layer fully connected perceptron consisting of 10 neurons was added to the VNNs with the same architecture as the ones that were trained on OASIS-3 dataset, we could improve the performance on the chronological prediction task. For 100 VNNs with adaptive readout that were trained on random permutations of the training data, the median MAE for the HC group was $4.64$ years, which was significantly smaller than the MAE achieved by VNNs with non-adaptive readouts (Section 4.1). Among the 100 VNN models with adaptive readouts, we analyzed the regional residuals for one VNN model with adaptive readout that had the best performance on chronological age prediction in HC group (test set: MAE = $4.17$ years, Pearson's correlation between prediction and ground truth = $0.73$; complete HC group: MAE = $4.26$ years, Pearson's correlation between prediction and ground truth = $0.725$). Our regional residuals revealed no significant difference between the regional residuals for AD+ group and HC group. This observation suggested that VNN lost its interpretability due to the addition of adaptive readout function. Moreover, we also observed a diminished gap between $\Delta$-Age for AD+ and HC groups determined using this VNN model ($\Delta$-Age for AD group: $1.58 \pm 4.67$ years, $\Delta$-Age for HC group: $0 \pm 3.45$ years, Cohen's $d = 0.384$). The findings discussed here suggest that boosting the performance on chronological age prediction task by using an adaptive readout function may penalize the interpretability offered by VNNs with non-adaptive readouts and also diminish the $\Delta$-Age gap between AD+ and HC groups.

## L   Results for randomly initialized VNNs

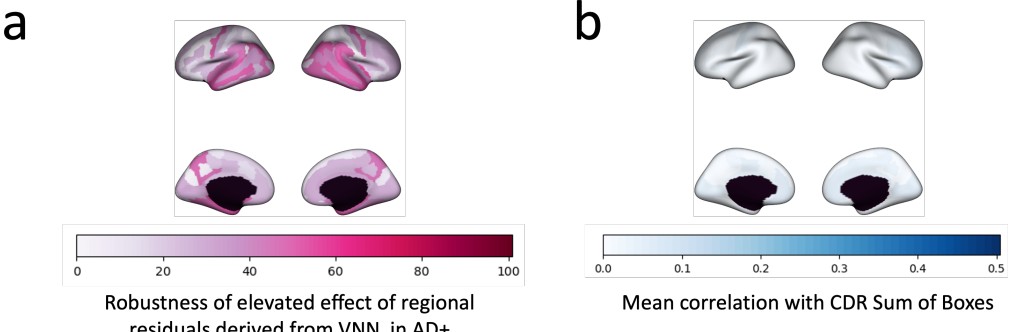

Figure 13: The results here were derived by VNN models that were randomly initialized and had the same architecture as those in Section 3.1.

## M Regional profiles corresponding to elevated regional residuals in AD+ group are stable to the composition of data used to estimate anatomical covariance matrix $C_{HA}$

Recall that $\Delta$-Age and associated regional profiles were evaluated using VNNs that operated upon a composite anatomical covariance matrix $\mathbf{C}_{HA}$. We next checked whether the results derived from VNNs relevant to $\Delta$-Age were stable to the changes in composition of the combined HC and AD+ groups used to estimate the anatomical covariance matrix. Note that the bilateral parahippocampal, entorhinal, subcallosal, and temporal pole regions are expected to be among the most relevant to $\Delta$-Age in AD based on the results in Fig. 3.

We performed two sets of experiments. In the first set, we included the whole HC group and gradually varied the number of individuals from the AD+ group to be included to form the estimate $\mathbf{C}_{HA}$. Figure 14a includes the results obtained from a randomly selected VNN model corresponding to the anatomical covariance matrix formed by different combinations of the individuals from HC and AD+ groups. The results in Fig. 14a display the brain regions whose regional residuals from AD+ group were higher than that in the HC group (Bonferroni corrected $p$-value $< 0.05$). The result obtained by the VNN when it used $\mathbf{C}_{HA}$ estimated from all $611$ HC individuals and $194$ AD+ individuals forms the baseline to evaluate stability in this context. When the covariance matrix $\mathbf{C}_{HA}$ was perturbed by using a smaller number of individuals from the AD+ group to estimate it, we observed that the significance of the relevant brain regions (parahippocampal, subcallosal and temporal pole) were preserved till exclusion of about $144$ AD+ individuals from the estimate $\mathbf{C}_{HA}$. Hence, the significant differences observed in the regional residuals for AD+ and HC groups in the aforementioned regions were robust to perturbations in $\mathbf{C}_{HA}$ due to variability in the number of individuals from the AD+ group.

Figure 14b illustrates the results obtained for a similar experiment as above, with the difference that the regional residuals were evaluated for the VNN when the anatomical covariance matrix $\mathbf{C}_{HA}$ was perturbed by reducing the number of individuals from the HC group used to estimate it. Using the result obtained for $\mathbf{C}_{HA}$ estimated from $611$ HC individuals and $194$ AD+ individuals in Fig. 14a as the baseline, the results pertaining to ANOVA between regional residuals for AD+ and HC groups (with AD+ elevated as compared to HC) remained consistent as long as $100$ or more individuals from HC group were included in forming $\mathbf{C}_{HA}$. With less than $100$ number of HC individuals included in $\mathbf{C}_{HA}$, the results became noticeably less significant in precuneus and supramarginal regions in the left hemisphere.

In summary, the group differences observed between the regional residuals for AD+ and HC groups in OASIS-3 dataset were robust to perturbations in the covariance matrix $\mathbf{C}_{HA}$ when it was perturbed from the baseline by using a different combination of HC and AD+ individuals to estimate it. However, we also remark that (nearly) complete exclusion of HC or AD+ groups from $\mathbf{C}_{HA}$ resulted in loss of significance of the elevation in regional residuals in AD+ for various regions, including bilateral parahippocampal and temporal pole regions, and precuneus and supramarginal regions in the left hemisphere. Thus, both HC and AD+ groups were relevant to the anatomical covariance matrix $\mathbf{C}_{HA}$ that resulted in regional profiles in Fig. 2a, and they were robust to the combination of individuals from HC and AD+ used to estimate $\mathbf{C}_{HA}$.

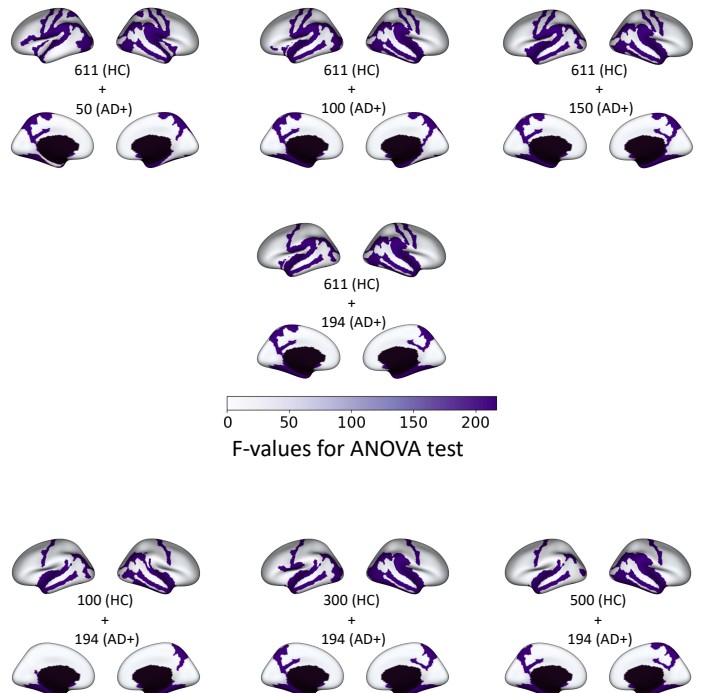

Figure 14: **Stability to perturbations in the anatomical covariance matrix for group differences between AD+ and HC groups observed in regional residuals.** For a VNN model that was trained to predict chronological age for HC group in OASIS-3 dataset, the regional residuals were first determined using the anatomical covariance matrix $\mathbf{C}_{HA}$ formed by the cortical thickness data of complete OASIS-3 dataset (i.e., 611 HC individuals and 194 individuals in the AD+ group. The group differences in regional residuals between AD+ and HC group were investigated according to the procedure in subsection 3.2. In the procedure described therein, we evaluated the F-values for the ANOVA test between regional residuals for AD+ group and HC group. The brain regions associated with the regional residuals that were significantly elevated in AD+ group with respect to HC group are highlighted on the brain template. The stability of the group differences to perturbations in $\mathbf{C}_{HA}$ was further investigated by varying the composition of cortical thickness data from AD+ and HC groups used to estimate $\mathbf{C}_{HA}$. Figures in the top row display the results obtained via analysis of regional residuals by VNNs that processes the cortical thickness data from the complete OASIS-3 dataset over the anatomical covariance matrix $\mathbf{C}_{HA}$ estimated from 611 HC individuals and a varying number of individuals from the AD+group. Figures in the bottom row illustrate the results of similar experiments, with the difference that the anatomical covariance matrix $\mathbf{C}_{HA}$ was estimated using all 194 individuals in the AD+ group and varying number of individuals from the HC group. The results corresponding to $\mathbf{C}_{HA}$ that was estimated using 194 AD+ individuals and 611 HC individuals formed the baseline for comparison for all scenarios.

# N  Δ-Age evaluation with anatomical covariance matrix from HC group

Using the anatomical covariance matrix derived only from the HC group (denoted by $\mathbf{C_H}$) resulted in observations that were consistent with Fig. 2 with a slightly diminished difference in Δ-Age between HC and AD+ groups. Specifically, Δ-Age for AD+ group in this setting was $3.41 \pm 4.57$ years, which was significantly larger than that for the HC group (ANOVA: partial $\eta^2 = 0.141$, Cohen's $d = 0.88$). The correlation between Δ-Age and CDR sum of boxes scores in the AD+ group was $0.464$. Thus, when the anatomical covariance matrix derived solely from the HC group was utilized in our brain age prediction framework, the magnitude of Δ-Age and its utility as a marker of dementia severity was slightly diminished as compared to the results in Fig. 2.

Figure 15a displays the regional profile associated with Δ-Age derived from VNNs with $\mathbf{C_H}$ as the anatomical covariance matrix. Comparison with Fig. 2a reveals that the robustness of regional residuals being elevated in the AD+ group in subcallosal and temporal regions was preserved, but that in bilateral entorhinal and parahippocampal regions was diminished in this scenario. Figure 15b displays the distributions for Δ-Age in AD+ and HC groups.

The variation in the regional residuals associated with entorhinal and parahippocampal regions in Fig. 15a and Fig. 2a could be explained by investigating the eigenvectors of $\mathbf{C_H}$. Specifically, Fig. 16a displays the associations between regional residuals for the AD+ group and the first 50 eigenvectors of $\mathbf{C_H}$, where we observed that the third, second, and first eigenvectors of $\mathbf{C_H}$ had the top three largest associations. Figure 16b plots the projections of the first three eigenvectors of $\mathbf{C_H}$ on a brain template. Note that the eigenvectors $\mathbf{v}_3$ and $\mathbf{v}_2$ have the largest weights associated with the subcallosal region in the right hemisphere, which is consistent with the relevant eigenvectors of $\mathbf{C_{HA}}$ in Fig. 3. However, unlike the eigenvectors of $\mathbf{C_H}$ in Fig. 16b, the eigenvectors of $\mathbf{C_{HA}}$ in Fig. 3b are also characterized by comparatively larger weights in the entorhinal and parahippocampal regions. Since parapippocampal and entorhinal regions are well known to be associated with disease onset and cortical atrophy [77], the anatomical covariance matrix $\mathbf{C_{HA}}$ may provide a more holistic perspective to Δ-Age in AD.

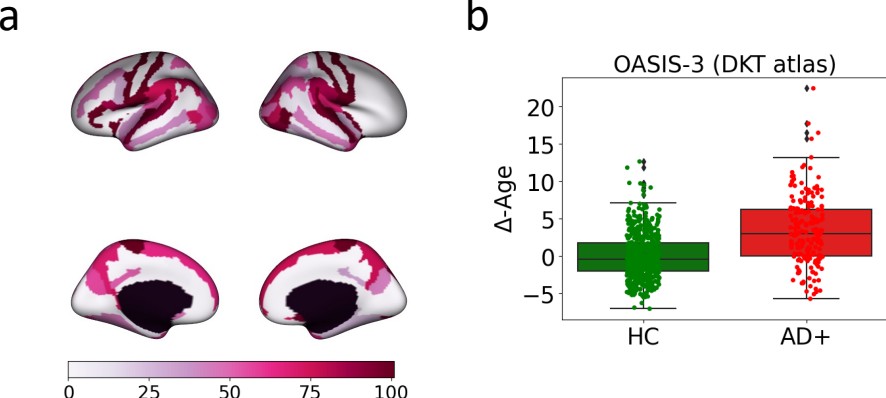

Figure 15: **Δ-Age results for anatomical covariance matrix from only HC group ($\mathbf{C_H}$).** Panel **a** projects the robustness of observing a significantly higher regional residual for AD+ group with respect to HC group for a brain region on the template. Panel **b** plots the Δ-Age distributions for AD+ and HC groups derived from VNNs with $\mathbf{C_H}$ as the covariance matrix.

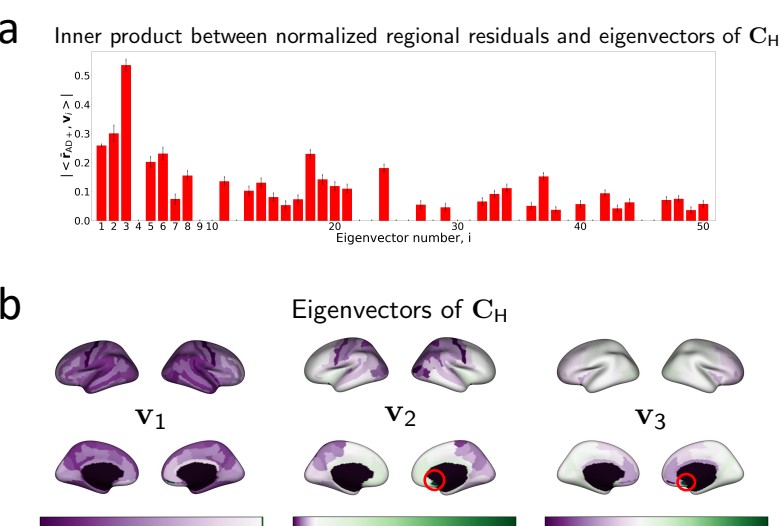

Figure 16: Panel **a** illustrates a bar plot for $|<\bar{\mathbf{r}}_{AD+}, \mathbf{v}_i>|$ for $i \in \{1, \ldots, 30\}$, where $\mathbf{v}_i$ is the $i$-th eigenvector of covariance matrix $\mathbf{C}_H$ associated with its $i$-largest eigenvalue. The bars are evaluated from the mean of $|<\bar{\mathbf{r}}_{AD+}, \mathbf{v}_i>|$ obtained for individuals in the AD+ group (results for eigenvectors associated with coefficient of variation of $|<\bar{\mathbf{r}}_{AD+}, \mathbf{v}_i>|$ larger than $30\%$ excluded). For every individual in AD+ group, the association of its regional residuals with eigenvectors of $\mathbf{C}_H$ were evaluated over $100$ nominal VNN models (trained on the OASIS-3 dataset). The eigenvectors associated with top three largest values for $|<\bar{\mathbf{r}}_{AD+}, \mathbf{v}_i>|$ are plotted on the brain template in Panel **b**. Subcallosal region in the right hemisphere was associated with the element with the largest magnitude in $\mathbf{v}_2$ and $\mathbf{v}_3$ and is highlighted with a red circle in the corresponding plots.

