# OpenReview forum: "Explainable Brain Age Prediction using coVariance Neural Networks"
_NeurIPS.cc/2023/Conference — NeurIPS 2023 poster_

### Official Review · Reviewer_HXvC · 2023-07-05

**Soundness:** 3 good
**Presentation:** 2 fair
**Contribution:** 2 fair
**Rating:** 5
**Confidence:** 4

**Summary:**

This paper proposes a new framework for predicting brain age, an area of increasing interest in computational neuroscience. The authors use coVariance Neural Networks (VNNs) to develop an anatomically interpretable method that relies on cortical thickness features. Their framework goes beyond existing metrics for the brain age gap in Alzheimer's disease (AD), revealing that VNNs can attribute anatomical interpretability to the brain age gap by identifying significant brain regions. It also demonstrates that this interpretability hinges on the VNNs' ability to leverage specific eigenvectors of the anatomical covariance matrix, offering an explainable approach to brain age prediction.

**Strengths:**

1. The paper addresses an important problem in Alzheimer's disease research by focusing on predicting the brain age gap. This is a critical aspect to understand and model, as it can potentially indicate an accelerated aging process due to adverse health conditions.
2. The proposed framework based on coVariance Neural Networks (VNNs) is a notable strength of the paper. Not only does it present a novel approach to brain age prediction, but it also offers improved interpretability, which is often lacking in complex neural network models.
3. The use of cortical thickness features for brain age prediction is well justified in the context of neuroscience. This choice is reasonable and lends biological plausibility to the models, likely improving their effectiveness and relevance to the task.

**Weaknesses:**

1. The learning rate selection for the Adam optimizer at 0.059 appears excessively specific and could be a sign of overfitting the hyperparameters to the data. The authors should provide an analysis or justification for this choice or consider testing a range of learning rates to demonstrate the robustness of the model to this parameter.
2. The paper's reliance on a single dataset with limited sample size is a notable weakness. Such a setting may limit the generalizability of the model and its results. The authors could improve this aspect by testing the model on additional datasets or considering methods to augment or diversify the existing data.
3. The lack of comparison with simpler or classic baseline models, such as linear regression, is another shortcoming. Such comparisons are necessary to demonstrate the model's superior performance and justify the additional complexity of the proposed approach. Without these comparisons, it is difficult to assess the true contribution of the paper's proposed method.

**Questions:**

Can you provide provide an analysis or justification for this choice of learning rate or consider testing a range of learning rates to demonstrate the robustness of the model to this parameter?

**Limitations:**

Yes

---

> ### Author Rebuttal · Authors · 2023-08-07
>
> Thank you for recognizing the novelty and key strengths of our work. We address the concerns raised in the review below.
>
> **Additional empirical evidence.**  To address the concern about empirical evidence, we have added results on the ADNI1 dataset (see pdf file attached with the global response). These results were derived from the models trained on OASIS-3 dataset and are highly consistent with the results reported in the paper.
>
> **Choice of learning rate.** The learning rate and other aspects of VNN architecture were chosen according to a hyperparameter search procedure on the training set using the package Optuna.
>
>
> The meaningful comparison with the studies in the existing literature could be categorized into the two following categories.
>
> **Comparison with traditional statistical models.** Since the chronological age prediction task using cortical thickness is a multivariate regression problem with correlated input features, a PCA-based regression model is the most appropriate traditional model for comparison with VNNs. However, this method obfuscates the anatomical interpretability of individual anatomic features as principal components could be linked with a combination of anatomical regions without any further insight. Importantly, PCA-based regression can be prone to instability due to small perturbations in the principal components and, hence, non-reproducible on settings with similar but perturbed principal components. Unlike PCA-based regression model, *VNNs offer theoretical stability guarantees on their performance that have been demonstrated empirically (see [a] and Appendix K in supplementary file)*.
>
> Other relevant approaches such as elastic-net regression require excessive fine-tuning of regularization parameters and could be overfit on the dataset characteristics (for e.g., the data processing pipeline used to extract cortical thickness features). Our results have demonstrated robustness to such aspects (for instance, VNNs trained on XYZ dataset processed according to ANTsCT pipeline and of dimensionality 100 could extract similar patterns of interpretability on OASIS-3 dataset that was processed according to Freesurfer; see Fig. 13 in the supplementary material).
>
> Furthermore, traditional statistical models operate within the dimensionality of a given dataset. In contrast, *VNNs are scale-free* and can process a dataset of arbitrary dimensionality. Hence, it is feasible for us to cross-validate the inference over datasets that may have different number of features. This is a relevant feature in neuroimaging and was utilized by us to demonstrate that anatomic interpretability results on OASIS-3 (148-dimensional) could be derived using VNN models that had been trained on XYZ dataset (100-dimensional). See Fig. 13 in supplementary file for details. We have further used this property of VNNs to cross-validate anatomical interpretability observed in OASIS-3 on an independent ADNI1 dataset (see **Fig. S2** in the attached pdf file with global response), where we observed consistent anatomical patterns for interpretability across datasets of dimensions 100, 200, and 400.
>
>
> **Comparison with deep learning methods with post-hoc explainability.** Lack of explainability is a well-recognized drawback of deep learning models. To address this, limited studies have utilized state-of-the-art post-hoc, model-agnostic methods such as SHAP or LIME [b] and saliency maps [c] to add explainability to their brain age estimation approaches, identifying the input features most relevant to the inference outcome. However, explainability offered by such post-hoc approaches may be unstable to small perturbations to the input, inconsistent to variations in training algorithms and model multiplicity (i.e., when multiple models with similar performance may exist but offer distinct explanations), and computationally expensive [d,e,f]. In this context, VNNs provide a transparent learning model that is inherently interpretable and can associate elevated brain age with brain regions characteristic of a disease or health condition as well as the principal components of the covariance matrix, with no significant added computational cost.
>
> **We will add the above discussion regarding comparison with existing methods in the literature to the literature review.**
>
> [a] Sihag, et. al, coVariance neural networks. In Proc. Conference on Neural Information Processing Systems, Nov. 2022.
>
> [b] A. Lombardi, et. al, “Explainable deep learning for personalized age prediction with brain morphology,” Frontiers in neuroscience, vol. 15, p. 578, 2021.
>
> [c] C. Yin, et. al, “Anatomically interpretable deep learning of brain age captures domain-specific cognitive impairment,” Proc. the National Academy of Sciences, vol. 120, no. 2, p.e2214634120, 2023
>
> [d] A. K. Dombrowski, et. al,  “Explanations can be manipulated and geometry is to blame,” Adv. in Neural Inf.Proc. systems, vol. 32, 2019.
>
> [e]  J. Adebayo, et. al, “Sanity checks for saliency maps,” Adv. in neural Inf.Proc. systems, vol. 31, 2018.
>
> [f] E. Black, et. al, “Model multiplicity: Opportunities, concerns, and solutions,” in Proc. the 2022 ACM Conference on Fairness, Accountability, and Transparency, 2022, pp. 850–863

---

> > ### Comment · Reviewer_HXvC · 2023-08-14
> >
> > Thanks for the detailed response. I think a table/figure to show the test performance change when varying the learning rate helps understand the method.

---

> > > ### Author Response · Authors · 2023-08-14
> > > **Varying learning rate**
> > >
> > > We thank the reviewer for their suggestion. We provide the following table to address reviewer's concern.
> > >
> > > Specifically, we investigated the results of chronological age prediction for the test set in HC group and brain age prediction in AD+ group in OASIS-3 dataset when the VNNs were trained with varying learning rates in the range [0.03, 0.2]. Similar to the results reported in the paper, we trained 100 VNN models for different permutations of the training and validation set and leveraged them to report the mean and standard deviation of the results for test performance as well as $\Delta$-Age for AD+ group. The results for learning rate 0.059 are also included to facilitate comparison.
> > >
> > >
> > > | Learning rate | Test set (MAE) | Test set (correlation) | $\Delta$-Age (AD+ group)|
> > >
> > > |_____ 0.03 ____ |_ 5.93$\pm$ 0.103_ |_____0.53$\pm$ 0.011 ___ | __ 3.33$\pm$ 4.22 ______ |
> > >
> > > |_____ 0.045 ___ |_ 5.96$\pm$ 0.19__ |_____0.53$\pm$ 0.019 ___ | __ 3.37$\pm$ 4.23 ______ |
> > >
> > > |_____ **0.059** ____ |_ **5.82$\pm$ 0.13**_ |_____**0.51$\pm$ 0.078** ___ | __ **3.54$\pm$ 4.49** ______ |
> > >
> > > |_____ 0.075 ____ |_ 5.98$\pm$ 0.24__ |_____0.54$\pm$ 0.018 ___ | __ 3.62$\pm$ 4.43 ______ |
> > >
> > > |_____ 0.10 _____ |_ 6.04$\pm$ 0.302_ |_____0.53$\pm$ 0.062 ___ | __ 3.58$\pm$ 4.31 ______ |
> > >
> > > |_____ 0.20 ____ |__ 6.26$\pm$ 0.407_ |_____0.54$\pm$ 0.08 ____ | __ 3.72$\pm$ 4.49 ______ |
> > >
> > > The results above demonstrate that the test performance diminishes slightly in terms of MAE when the learning rate is smaller than 0.059. With learning rates 0.1 or 0.2, we observed limited but relatively more significant degradation in MAE (coupled with increased variance across 100 VNN models). However, for all scenarios, the mean of the Pearson's correlation between the VNN outputs and ground truth was consistently above 0.5. Also, for all scenarios, the $\Delta$-Age estimated by VNNs for AD+ group was consistently elevated, as has been reported in the paper.
> > >
> > > Hence, **the above results demonstrate that the findings reported in the paper were robust to choice of learning rate within a reasonable range**. We are happy to address any further concerns the reviewer may have.

---

> > > > ### Comment · Reviewer_HXvC · 2023-08-18
> > > >
> > > > Thanks for your reply. I will increase my rating.

---

> > > > > ### Author Response · Authors · 2023-08-18
> > > > >
> > > > > Thanks again for your comments and for raising your recommendation of our work.

---

### Official Review · Reviewer_n5r3 · 2023-07-05

**Soundness:** 3 good
**Presentation:** 3 good
**Contribution:** 2 fair
**Rating:** 4
**Confidence:** 4

**Summary:**

The authors propose a framework for predicting brain age by developing coVariance Neural Networks (VNN). They leverage the stability properties of VNN to first train the network using data from healthy controls to predict chronological age. Then, they perform inference using a combined dataset that includes groups with Alzheimer's disease (AD+), using the covariance matrix of the data. On the regional residuals, they conduct statistical analysis to identify distinct brain regions affected by the disease and demonstrate a strong correlation between the residuals and specific eigenvectors, suggesting that the framework enables anatomical interpretability. By utilizing a simple linear model to correct for age bias, the authors obtain the brain age after predicting the chronological age. Furthermore, the authors provide additional discussions on eigenvector(s) and their implications.

**Strengths:**

- This paper is well-organized and effectively motivated, providing a clear and comprehensive explanation. The suitability of VNN for the target task is convincingly demonstrated.
- The framework presented in this paper takes advantage of the covariance structure, which is further supported by eigenvector studies. This approach allows for a reduction in the number of learnable parameters, making it particularly suitable for the medical domain where sample sizes are often limited.


**Weaknesses:**

- This paper lacks original contribution, as it primarily builds upon existing studies with some additional adaptations and explanations for the specific task.
- The rationale for training 100 VNNs is not explained properly. It is unclear whether this large number is necessary to overcome any deficiencies or solely for the benefits of ensemble learning. Furthermore, the inferior performance compared to other approaches when using the additional ensemble method should be properly addressed.
- Relatively large residuals in specific regions may indicate the impact of a particular disease, and this is perhaps the reason why the authors used ANOVA for detailed diagnosis. However, while higher correlation between residuals and specific eigenvectors may demonstrate validity, it does not necessarily imply interpretability.
- Discussions on scalar parameters (filter taps) would be beneficial as they provide how much to utilize specific-sized neighborhoods.


**Questions:**

- Does replacing CHC with CHA result in significantly improved outcomes?
- Is it still feasible to do so even when there are a large number of input variables? Is there a minimum F value that corresponds to the number of variables?


**Limitations:**

Authors adequately discuss their limitations.

---

> ### Author Rebuttal · Authors · 2023-08-07
>
> Thank you for recognizing the appropriateness of VNNs to the brain age prediction task and broadly to data analysis in medical domain. The concerns in the review are addressed below.
> ### Contributions and novelty.
> The motivation to study VNNs for brain age prediction relative to other studies in brain age prediction hinges on the following aspect.
>
> **Fallacies of focusing on performance.** Most existing brain age prediction studies use the performance on the chronological age prediction as a prominent metric to assess the quality of brain age. However, the performance on chronological age prediction is a flawed metric for assessing the quality of brain age estimate. Particularly, in the absence of explainability, the performance solely cannot provide clarity on the following aspects:
> - Does better performance on predicting chronological age correlate with a more *useful* estimate of brain age?
> - Are all models that achieve a specific mean prediction error (say 1 year) on the chronological age prediction task *equivalent* in terms of their ability to predict a meaningful brain age in adverse health conditions?
>
> A recent study of several existing brain age prediction frameworks has revealed that the accuracy achieved on the chronological age prediction task may not correlate with their clinical utility (see [a]). Further, the age-bias correction step in brain age evaluation procedure accounts for any inaccuracy in the chronological age prediction (whether the Pearson’s correlation between estimated chronological age and ground truth is 0.9 or 0.5). *These observations suggest that explainability must be the key metric to assess the biological plausibility of a brain age algorithm, irrespective of its performance on the chronological age prediction task.* The interpretability offered by VNNs facilitates a convincing evaluation of the biological plausibility of brain age estimates and has not been explored before.
>
> **Conceptual contributions.**  Furthermore, *VNNs provide methodological clarity to all aspects of brain age prediction*. Specifically, VNNs learning 'something significant' from the chronological age data as regression models is a necessary first step for attaining robust anatomical interpretability (Appendix H.2 in supplementary file provides more evidence in this context). This interpretability hinges on exploitation of specific eigenvectors of the anatomical covariance matrix. The VNNs learn to exploit these eigenvectors when trained on the task of chronological age prediction. Further, the age-bias correction step is limited to projecting the VNN output onto a space where brain age can be compared meaningfully to chronological age. Due to word limit here, we also refer the reviewer to comparison with existing approaches in the response to Reviewer mFGa.
>
> **[a]** Jirsaraie, et. al, A systematic review of multimodal brain age studies: Uncovering a divergence between model accuracy and utility. Patterns, 2023.
>
> **We will incorporate the above discussion to better communicate the contributions of VNNs to brain age prediction task.**
>
> ### Training on 100 VNN models.
> We report the performance derived from *100 VNN models to demonstrate the high confidence in our findings*, i.e., our findings were not a product of only one model. Existing literature shows that it is possible to have different explainability for deep learning models with similar performance. Such an observation can reduce the confidence in explainability offered by only one model. In this context, our findings in Fig. 2a were highly consistent across 100 VNN models, thus, suggesting that the results were not overfit on a specific training set.
>
> ### Eigenvectors and interpretability.
> First, we refer the reviewer to Fig. 9 in the supplementary file, which plots three eigenvectors with the largest associations with regional residuals on a brain surface. The comparison of anatomical interpretability in Fig. 2a and eigenvector plots in Fig. 9 suggests that the VNN’s ability to exploit these eigenvectors was instrumental to recovering the results in Fig. 2a. This observation is indeed verified by our results in Fig. 13 in supplementary file where randomly initialized VNNs are unable to provide robust patterns of anatomic interpretability. Hence, the ability of VNNs to exploit these eigenvectors is instrumental to the observed anatomical interpretability.
>
> ### Filter taps in VNNs.
> Since the first layer consisted of 5 filter taps and the second layer consisted of 10 filter taps, the overall neighborhood size is 13 (as the first filter tap in each layer is not tied to the convolution operation).
>
> ### Number of input variables and anatomical interpretability.
> The number of input variables can vary widely across neuroimaging datasets. Currently, we assess the significance using p-values in ANOVA obtained after Bonferroni correction which provides consistent results for datasets with 148, 100, 200, and 400 number of features. However, datasets with a larger number of features also provide more localized results as compared to datasets with smaller number of features. A minimum F-value corresponding to number of features may provide a more accurate mechanism to assess the significance of our results. We will discuss this aspect in the limitations of our analysis and potential future work.
>
> ### Choice of covariance matrix.
> There was no statistically significant difference in brain age gaps derived for the AD+ group for the two choices of covariance matrices (results for ${\bf C}_H$ are provided in Appendix L). In terms of interpretability, we observed reduced significance for certain regions when ${\bf C}_H$ was used.  Since there are no current ground truths or benchmarks to evaluate interpretability, studying the associations of brain age estimates with various clinical and genetic markers of dementia is warranted to better understand the biological plausibility of the two choices.

---

> > ### Comment · Reviewer_n5r3 · 2023-08-18
> >
> > I appreciate the authors for their clarification. I believe that this paper provide good novelty from cognitive science perspective, but I am still quite concerned if the finding given in this paper is robust. It is quite unfortunate that no baseline was provided, therefore it is still hard to convince myself what the gain is. The authors strongly argue that all previous approaches focus on predicting chronological age prediction which is a false metric, and thus it would have been nice if new findings from this work over the previous approaches had been presented side by side. I agree with many of the points that Rev m6iX has raised, and I would like to keep my score for now.

---

> > > ### Author Response · Authors · 2023-08-19
> > > **Revised response to Reviewer n5r3**
> > >
> > > *Note: After further discussion among the authors, we have expanded our response to Reviewer n5r3. The previous response is included and the revisions are included under the **Edit** section.*
> > >
> > > We thank the reviewer for their acknowledgement of the key arguments in our response and appreciate their feedback.
> > >
> > > Our claims of robustness rely on our observation that 100 distinctly trained VNN models isolated certain brain regions as contributors to elevated brain age gap consistently on multiple datasets of distinct dimensionalities.
> > >
> > > We also clarify that our arguments pertain to the insufficiency of the performance of chronological age prediction as a metric for assessing the quality of brain age estimate, while not discounting the relevance of chronological age in this application. More specifically, the model must learn the information about healthy aging from chronological age data but the performance achieved on this task is an incomplete metric to assess whether the model is able to provide a biologically plausible brain age estimate in neurodegeneration.
> > >
> > > **Edit.**
> > > We understand the reviewer's concern regarding the robustness of our results, which we believe to focus on investigating whether our results are spurious -- or "robust" as the reviewer states. Our experiments were very much focused to tackle this aspect convincingly and hence, we believe that we have more common ground with the reviewer than it seems.  Where we depart from the reviewer is in the use of baselines to establish this robustness. As much as we would like to provide these comparisons, we can't think of any comparison that would be fair and meaningful.
> > >
> > > Rather, we attempt to corroborate robustness by **interpretability** and **generalization**. Indeed, the connection between VNNs and eigenvectors of anatomical covariance matrix allows us to identify brain regions that are responsible for elevated brain age. These regions turn out to have clinical significance. And we have further demonstrated that a VNN that we train on one specific dataset generalizes to different resolutions and different datasets. *This is a strong indication that the features we are extracting are not spurious.*
> > >
> > > This is perhaps not **the** way in which the reviewer would like us to show that our results are not spurious but is certainly **a** way of showing so. At the very least, we think that we can all agree that, in the words of Reviewer m6iX, *"there is a message here that will be of interest to the field."*

---

### Official Review · Reviewer_m6iX · 2023-07-06

**Soundness:** 3 good
**Presentation:** 2 fair
**Contribution:** 2 fair
**Rating:** 5
**Confidence:** 3

**Summary:**

The authors aim is to investigate the application of coVariance Neural Networks (VNNs) to brain age prediction. They train and test VNNs on the OASIS-3 brain dataset with additional data of Alzheimer's disease and cortical thickness. The results indicate an association between the biomarker (brain age prediction) and a more established biomarker (cortical thickness).

**Strengths:**

In this paper, the authors approach an interesting problem with many practical applications: interpretable brain age predictions. The paper is well written and organised, making it accessible to a broad audience. The introduction effectively describes the problem and highlights its significance, while the literature review provides a broad view of the field and related fields.

The methodology section is detailed, providing a clear explanation of the techniques used and has a deep description of the previously-published approaches used, which is helpful for the reader. It might have benefitted the paper to see more thorough discussion of the rationale behind the chosen methods. The discussion provides a good summary of the research findings and their implications. (However, these sections could be expanded to include a more comprehensive discussion of the potential applications and limitations of the study, as well as directions for future research).

**Weaknesses:**

Alongside the strengths discussed above, there are a number of weaknesses that must also be acknowledged or addressed. Notably:
- A significant portion of the paper is devoted to introducing and discussing VNNs, which have already been published
- There is a lack of references and discussion to previously published methods for interpreting brain age predictions. [1, 2, 3] just to name a few examples. How does this method differ from previous approaches? Does this method corroborate or pick up new regions of interest? The second reference [2] appears in the list of references in the paper but it not discussed anywhere.
- Lack of results from the contributing brain regions. I would have liked to see the a table or figure with all brain regions and their associations. The VNN section can be reduced to create space for this.
- The applications might not be directly obvious to those outside the field or on the periphery. A conclusion paragraph to discuss the impact of this work and future steps would help place it the current state of the world.

[1] Hofmann, S. M., Beyer, F., Lapuschkin, S., Goltermann, O., Loeffler, M., Müller, K. R., ... & Witte, A. V. (2022). Towards the interpretability of deep learning models for multi-modal neuroimaging: Finding structural changes of the ageing brain. NeuroImage, 261, 119504.

[2] Lee, J., Burkett, B. J., Min, H. K., Senjem, M. L., Lundt, E. S., Botha, H., ... & Jones, D. T. (2022). Deep learning-based brain age prediction in normal aging and dementia. Nature Aging, 2(5), 412-424.

[3] Kolbeinsson, A., Filippi, S., Panagakis, Y., Matthews, P. M., Elliott, P., Dehghan, A., & Tzoulaki, I. (2020). Accelerated MRI-predicted brain ageing and its associations with cardiometabolic and brain disorders. Scientific Reports, 10(1), 19940.

**Questions:**

- How is the alignment of images done? Could bias be introduced here?
- Is the age-correction in 3.3 performed on the train or test group? It sounds like the parameters are obtained from the test group, which can lead to overfitting
- In line 42-43 appears the sentence: "Thus, there is a lack of conceptual clarity in the role of training to predict chronological age of healthy controls in predicting a meaningful ∆-age [19].". How does this statement reflect the work being performed here? Surely there are points of discussion worth addressing.
- Why train cortical thickness only on healthy group? Particularly in light of the previous question.

**Limitations:**

Limitation discussion is more a discussion on the general field of brain age predictions rather than the specific limitations of this approach. It does not demonstrate the authors’ awareness of the limitations of the analysis.

There is no discussion on or consideration made to ethical concerns. Although this work has many benevolent applications, any system that can predict a person’s age, or interpret the prediction in an anatomic way, can be abused in potentially unethical or dubious ways. Using the output to refuse refugee applications or increase insurance premiums being two examples. Although I don't think a separate ethics review is needed, I encourage the authors to show awareness of possible misuse of their work.

---

> ### Author Rebuttal · Authors · 2023-08-07
>
> ### Novelty, conceptual contributions, and comparisons with existing methods.
> Here, we clarify the lack of conceptual clarity associated with chronological age in brain age prediction, motivation for using VNNs, and the comparisons with existing brain age approaches. These aspects are intertwined and, hence, responded to jointly.
>
> **Limitations of 'performance'.** Many existing brain age algorithms primarily focus primarily on the performance in chronological age prediction task. However, improved performance over chronological age prediction task cannot ensure improved biological plausibility of the brain age estimate (see [a] below). Hence, it’s imperative to have explainability of the brain age estimate as the key metric for assessment of any algorithm in this domain.
>
> **Choice of VNNs for brain age prediction.** To start with, the interpretability offered by VNNs facilitates a convincing evaluation of the biological plausibility of brain age estimates. Furthermore, *VNNs provide methodological clarity to all aspects of brain age prediction*. Specifically, VNNs learning 'something significant' from the chronological age data as regression models is a necessary first step for attaining robust anatomical interpretability. This interpretability hinges on exploitation of specific eigenvectors of the anatomical covariance matrix. The VNNs learn to exploit these eigenvectors when trained on the task of chronological age prediction. Further, the age-bias correction step is limited to projecting the VNN output onto a space where brain age can be compared meaningfully to chronological age.
>
> **Comparison with existing approaches.**  The studies suggested by the reviewer broadly fit into the category of deep learning methods with post-hoc explainability and will be added to the literature review. We note that such approaches do not rigorously account for the limitations of post-hoc explainability, some of which include: (i) instability to small perturbations to the input, (ii) inconsistent results for different variations of training algorithms, and  (iii) model multiplicity (i.e., when multiple models with similar performance may exist but offer distinct explanations). In contrast, VNNs provide a transparent learning model with no added computational cost for explainability.
>
> **Interpretability offered by VNNs.** Our analysis identified the following regions as prominent contributors to elevated brain age gap in AD: entorhinal, superior temporal, temporal pole, and subcallosal. Among these, entorhinal is implicated in earlier stages of AD according to Braak staging criteria, and others implicated in later stages of AD. For instance, the study in [b] implicates regions in temporal lobe among prominent contributors to brain age in MCI subjects.
>
> **[a]** Jirsaraie, et. al, A systematic review of multimodal brain age studies: Uncovering a divergence between model accuracy and utility. Patterns, 4(4), 2023.
>
> **[b]** Ran, Chen, et al. "Brain age vector: A measure of brain aging with enhanced neurodegenerative disorder specificity." Human brain mapping, 2022.
> ### Training on healthy group.
> A key biological feature of AD is manifestation of biological characteristics that signify accelerated aging relative to healthy aging. Hence, the models were trained to learn characteristics of healthy aging and deployed on the AD cohort to detect accelerated aging.
> ### Age-correction and additional empirical evidence.
> Indeed, the models were trained on the HC group, although the AD+ group was unseen. To address the concern regarding overfitting on HC group and further provide evidence of the generalization of our results, we leveraged the models trained on OASIS-3 to predict brain age in ADNI1 dataset (demographics and results included in the pdf file attached with the global response). We observed larger brain age gap in dementia with similar brain regions being implicated as in the results on the OASIS-3 dataset.
> ### Impact and future work.
> An immediate future direction is to explore the associations of brain age gap and regional residuals with clinical and biological markers of AD. Utility of other cortical features (such as volume or area) in brain age prediction can also be explored. Furthermore, the VNN based explainability framework is also a potentially impactful data analysis approach. Specifically, we have demonstrated the connection between the inference outcome and eigenvectors of the underlying covariance matrix while validating the key property of stability and reproducibility of findings.
> ### Limitations of analysis.
> Our analysis is limited to older individuals, and a dataset with more diverse age groups is expected to provide holistic information for brain age. Isolation of brain regions contributing more to brain age in AD than HC hinges on a binary group comparison. Such a comparison can be impacted by the composition of the dataset (for instance, a skewed dataset may not provide informative results). We will discuss this aspect in the limitations section.
>
> ### Results from the contributing brain regions.
> Fig. 11 in the supplementary material provides a subset of this data for one model. We will compile this data from all 100 VNN models in the supplementary file.
> ### Description of VNNs.
> As per the suggestion, we will optimize this section further by moving some content to the supplementary file.
> ### Alignment of images.
> OASIS-3 dataset was processed via Freesurfer pipeline, and we use the derived cortical thickness data available online in the data repository. The relevant processing details are provided in the data dictionary document for this dataset (available online). A pertinent detail therein is that all FreeSurfer outputs were quality checked for errors in segmentation before upload. Hence, we did not expect the bias due to alignment in our results.
> ### Societal concerns.
> Thank you for raising the relevant societal concerns. We will add these to our discussion.

---

> > ### Comment · Reviewer_m6iX · 2023-08-18
> > **Reply to authors**
> >
> > I thank the reviewers for taking the time to reply to my review and for taking it into consideration.
> >
> > I am conflicted about the work. The authors have answered many of my questions but I still would have liked to see more connections to existing work on a discussion level. Explainability can be enigmatic and a quantitative comparison with other methods might not be meaningful.
> >
> > There are also minor things that were left unaddressed such as works listed in the references not being cited anywhere in the paper (I suspect this might be because the authors copied the references over from the supplementary but I would suggest keeping them separate for clarity).
> >
> > Having said that there is a message here that will be of interest to the field and the additions of related work and improved discussion on limitations improves its value. Therefore I am prepared to improve my rating from 4 to 5.

---

> > > ### Author Response · Authors · 2023-08-18
> > >
> > > We thank the reviewer for their feedback and are encouraged by their appreciation of the message of our work. Certain references were indeed a part of the Relevant Literature section in the supplementary material. We will integrate the reviewer's suggestions on the inclusion of more comprehensive discussions relative to existing works and better communication of the limitations.

---

### Official Review · Reviewer_mFGa · 2023-07-06

**Soundness:** 4 excellent
**Presentation:** 4 excellent
**Contribution:** 4 excellent
**Rating:** 7
**Confidence:** 4

**Summary:**

The paper leverages coVariance neural networks (VNNs) for brain age gap prediction in a principled statistical fashion. The paper focuses on the specific case of training on a healthy control and evaluating the gap on people with Alzheimer's disease.

------------------------------------
EDIT AFTER REBUTTAL PERIOD:
I will increase the score I gave to this paper from 6 to 7 (accept), and increase the Soundness and Contribution scores from 3 to 4. I'm more confident about the relevance of this work compared to when I first reviewed it, and I hope the remaining reviewers can engage in this discussion too. We still seem to disagree on the need for performance-based baselines.

**Strengths:**

The paper is well written and organised. In contrast to other relevant methods in the field that need post-hoc approaches for explainability, this paper is able to offer a transparent statistical approach given the regional expressivity of the VNN architecture. This is a key and very relevant strength of the paper, given the gap in the literature and the need for interpretable methods if one really wants to have useful machine learning applications in healthcare practice. It is particularly good that the paper demonstrates that the age gap differences between AD+ and HC groups were not driven by age or sex differences.

**Weaknesses:**

1. I think the paper lacks a comparison of their method with other baselines. I understand that the contribution is about the principled statistical method, but without a comparison with previous methods it is difficult to understand how good the method is at least for the dataset analysed. It is highlighted in section 4.1 that other DL methods achieve better MAEs; how much is this difference in the case analysed in this paper? If the difference is too high, the interpretability advantages of a method are lost in practice, because if the method is way worse in predicting age gap, it is of lesser importance that we can interpret the results. I am aware, as the paper defends, that a very accurate method might not exactly be the best, but this information is important for a proper contribution analysis with the tools that we have available.
2. The fact that this paper only uses one main single dataset to evaluate their method is a key weakness. As I further ask in the Questions section, why haven't the authors used another dataset, like the Human Connectome Project (HCP), for a more wider evaluation of the relevance of this work?
3. The Pearson's correlation achieved (as stated in section 4.1) is very close to 0.5 which raises the question of how much the model actually learned. Furthermore, the MAE of above 5 seems to be in the quantile range of figure 2b, thus being in the range of where most of the values are actually predicted anyway.
4. Small typo in Discussion section: "near-prefect".


**Questions:**

1. I'm finding a bit confusing to evaluate the contribution of this paper given that VNNs were previously presented in literature, and therefore I get the impression that the actual contribution is more about the added explainability analysis; however, most part of section 4 is about the results of the VNNs not necessarily taking into account the interpretable part. Can the authors more clearly explain the differences of this work to previously literature?
2. In section 2.2 the paper states that a particular useful characteristic of VNNs is that they can process a dataset of an arbitrary dimensionality. Wouldn't it be useful then to show the method applied to different datasets with a different number of features? Or, even just the same dataset in which the number of features are different?
3. I question the choice of healthy control dataset to train the VNN. The OASIS-3 dataset has a mean age of 68 years old, which might raise the question of how "healthy" this control actually is. Why didn't the authors use another dataset with a healthier cohort, like the Human Connectome Project (HCP)?
4. Is there any particular reason for the authors to only focus on cortical thicknesses for age prediction? Have the authors tried to include other measures (eg volume/area) and see whether the model performance increases?


**Limitations:**

Limitations of this work are stated in the paper at different places, including at the end. No potential negative societal impact of this work seems to have been discussed. I'm thinking that a wrong prediction of a brain age gap (given implications for neurodegenerative diseases) could potentially imply over-medication in the case where the brain age gap predicted is wider than reality, as just one example.

---

> ### Author Rebuttal · Authors · 2023-08-07
>
> ### Contribution and novelty.
> To understand the contributions relative to other studies in brain age prediction application, the following aspect must be recognized explicitly.
>
> **Fallacies of focusing on performance.** The performance on chronological age prediction is a flawed metric for assessing the quality of brain age estimate, as it cannot provide clarity on the following aspects:
> - Does better performance on predicting chronological age correlate with a more *useful* estimate of brain age?
> - Are all models that achieve a specific mean prediction error (say 1 year) on the chronological age prediction task *equivalent* in terms of their ability to predict a meaningful brain age in adverse health conditions?
>
> A recent study of several existing brain age prediction frameworks has revealed that the accuracy achieved on the chronological age prediction task may not correlate with their clinical utility (see [a]). Further, the age-bias correction step in brain age evaluation procedure accounts for any inaccuracy in the chronological age prediction (whether the Pearson’s correlation between estimated chronological age and ground truth is 0.9 or 0.5). (Here, we remark that we do not advocate for an arbitrarily suboptimal training of a given model on the chronological age prediction task.) These observations enable the question: What is the appropriate metric to assess validity of brain age prediction algorithms?
>
> **Conceptual contributions.**  To start with, the interpretability offered by VNNs facilitates a convincing evaluation of the biological plausibility of brain age estimates. Furthermore, *VNNs provide methodological clarity to all aspects of brain age prediction*. Specifically, VNNs learning 'something significant' from the chronological age data as regression models is a necessary first step for attaining robust anatomical interpretability (Appendix H.2 in supplementary file). This interpretability hinges on exploitation of specific eigenvectors of the anatomical covariance matrix. Notably, the VNNs learn to exploit these eigenvectors when trained on the task of chronological age prediction (Fig. 13 in supplementary file). Further, the age-bias correction step is limited to projecting the VNN output onto a space where brain age can be compared meaningfully to chronological age.
>
> **We will incorporate the above discussion and replace Fig. 3 in the main paper with Fig. 9 from supplementary material to better communicate the explainability feature of VNNs.**
>
>
>
> **Comparison with existing approaches.** The meaningful comparison with existing approaches is provided as follows.
> 1. *Comparison with deep learning methods with post-hoc explainability.*  Limited studies have utilized state-of-the-art post-hoc, model-agnostic methods such as SHAP or LIME and saliency maps to add explainability to their brain age estimation approaches. However, explainability offered by such approaches may be unstable to small perturbations in data, inconsistent to variations in training algorithms and model multiplicity (i.e., when multiple models with similar performance may exist but offer distinct explanations), and computationally expensive [b,c,d]. In this context, interpretability is an inherent feature of VNNs, which comes with no significant computational cost.
> 2. *Comparison with traditional statistical models.* A PCA-based regression model is a standard method and one of the most appropriate traditional model for comparison. However, this method obfuscates the anatomical interpretability of individual anatomic features. Importantly, it can be prone to instability due to small perturbations in the principal components and, hence, non-reproducible. Unlike this statistical model, VNNs offer theoretical stability guarantees on their performance that have been demonstrated empirically (see [e] and Appendix K).
> 3.  *Baseline performance by VNNs with perceptron as readout.* We can, of course, artificially improve the VNN performance in chronological age prediction task further by the use of adaptive readout layer in VNNs (see Appendix J in supplementary file; where error of 4.17 years was achieved by VNNs with a perceptron as a readout layer). However, this modification inhibits the anatomical interpretability and the scale-free property of VNNs.
>
> [a] Jirsaraie, et. al, A systematic review of multimodal brain age studies: Uncovering a divergence between model accuracy and utility. Patterns, 4(4), 2023.
>
> [b] A.-K. Dombrowski, et. al,  “Explanations can be manipulated and geometry is to blame,” NeurIPS, 2019.
>
> [c]  J. Adebayo, et. al, “Sanity checks for saliency maps,” NeurIPS, 2018.
>
> [d] E. Black, et. al, “Model multiplicity: Opportunities, concerns, and solutions,” ACM Conf. on Fairness, Accountability, and Transparency, 2022.
>
> [e] Sihag, et. al, coVariance neural networks. NeurIPS 2022.
> ### Additional empirical evidence.
> We have cross-validated the findings on another dataset (ADNI1; see global response).
> ### Choice of dataset.
> Datasets in AD studies typically focus on older healthy controls who had been clinically screened. In both OASIS-3 and ADNI, controls were age-matched with the respective AD cohorts. In principle, a dataset that represents the adult lifespan may provide more holistic information about healthy aging. Other choices of features, such as cortical area and volume, can certainly provide more insights into brain age and VNNs are indeed applicable to them. We will mention these aspects as immediate future directions of our work.
> ### Scale-free property of VNNs.
> The suggestion to show the VNNs applied to datasets with different number of features is highly relevant. In **Fig. S2** in the pdf file attached with the global response, we demonstrate the transferability of VNNs, where we report consistent interpretability patterns for brain age on datasets of different dimensionalities.
> ### Limitations.
> Thanks for this suggestion. We will add it in Limitations section.

---

> > ### Comment · Reviewer_mFGa · 2023-08-12
> >
> > I thank the authors for the detailed answers to my review, and for the global answer to all reviewers. The experiments on the extra dataset are very very good to illustrate the relevance of this paper. I agree with the highlighted advantages of the paper that the authors wrote in their rebuttal, but I believe some of my points were still not sufficiently tackled.
> >
> > With regards to the weaknesses that I defined, I still think that the lack of comparison with baselines is a weakness of this work. I think I understand that current metrics of performance for brain age gap prediction might not be the best, but it's still a common practice in the literature. The fact that the authors are only able to cite one previous paper to support this decision illustrates my point: yes, ref [a] shows that we might need to rethink how we do research in this area, but I still believe that information on current available metrics is important for a proper analysis with the tools that we have available. If there's only one previous work highlighting this issue I don't think the correct approach is to blindly follow it. I argue that the correct approach for a high-impact paper should be to show experiments and connect with this previous work. In the wider context of the literature, I argue that showing these comparisons will give a wider picture of how your method compare with previous ones and how they can relate with ref [a]'s concerns. In this sense, the fact that the Pearson's correlation achieved (as stated in section 4.1) is very close to 0.5 still raises the question of how much the model actually learned. I understand that this might be a suboptimal metric, but isn't it a bad indication that it's so close to 0.5? Being a suboptimal metric doesn't mean it doesn't illustrate some information for us to understand the model.
> >
> >
> > It also seems to me that the authors did not answer my questions 1 and 4. For question 1, I'm a bit confused about the contributions of this paper given that VNNs were introduced in previous work, so I'd like to ask for a clarification on the different contributions of this work when compared to the previous ones. For question 4, I understand that authors defend their method can be easily extended, but I was trying to look for the reasoning that the authors had for choosing only cortical thicknesses. I'm sorry if I missed this in your answers, but these questions still seem not answered to me.
> >
> >
> > I look forward to the discussion with the other reviewers, as I think they raised important points and would like to know whether they think the authors sufficiently tackled their concerns.

---

> > > ### Author Response · Authors · 2023-08-12
> > > **Follow-up to Reviewer mFGa's comments**
> > >
> > > We thank the reviewer for their valuable feedback to our rebuttal. Our discussion below first addresses Q.1 of the reviewer, where we provide further clarifications on how our work is positioned relative to the existing works in brain age prediction domain, while also addressing their comments regarding the relevance of performance.
> > >
> > > We hope that the reviewer will agree that an elevated brain age for an individual with neurodegeneration, by itself, does not add much clinical utility beyond validating what a clinician can already observe using a variety of other biomarkers (for e.g., NfL). The focus on ‘chronological age prediction performance’ is indeed a common practice for the domain of machine learning (ML) driven brain age prediction, and many existing algorithms already show elevated brain age gaps for a variety of phenotypes. Existing literature provides sufficient evidence of many sophisticated deep learning models, with thousands to millions of learnable parameters, that can readily achieve a very high accuracy on the regression task of predicting chronological age. In this context, *we argue there is no significant methodological innovation to be made in demonstrating accurate predictions for chronological age$^1$*.
> > >
> > > **To put it simply, a key focus of this paper is not on the accuracy in predicting chronological age, but rather ‘what properties does a VNN gain when it is exposed to the information provided by chronological age of healthy controls’ and ‘whether these properties lead to an informative brain age estimate in AD’.** While highly relevant, most existing studies fail to tackle these aspects convincingly or even consider them. Our experiments have demonstrated that the information gleaned by VNNs from chronological age of healthy controls$^2$ is sufficient to estimate brain age in AD that is biologically plausible (as it isolates the brain regions characteristic of AD as contributors to elevated brain age gap). *Hence, we provide a methodologically holistic perspective to brain age prediction task, to which we do not find any appropriate benchmarks to compare to in the existing literature.* It could be the case that improving the performance on chronological age prediction beyond that achieved by VNNs indeed improves this biological plausibility in some form. However, there has not been appropriate focus on this aspect in the literature and to this end, we provide a potential benchmark on how to investigate this from a methodological perspective for future work in this domain.
> > >
> > > The reviewer rightly points out the sparsity in literature that argues for a decoupling of brain age from performance on chronological age prediction. Besides the study [a], a previous study (see Bashyam et al. published in Brain, 2020) also showed evidence of moderately fitted models on chronological age achieving a more informative brain age on a large dataset. However, a 'moderate' fit is hard to define concretely when the model can achieve a much improved performance on the task it is being trained for. *The skewed focus on performance on chronological age prediction task in the brain age domain is a by-product of the lack of ‘conclusively’ explainable models$^3$*. We hope that our work will help bring appropriate attention to the explainability of brain age prediction models, as it is paramount for the practical utility of ML derived brain age estimates.
> > >
> > > **We hope that the discussion above answers Q.1 raised by the reviewer as we have argued that the major contributions of our work are both conceptual and methodological, where we not only bring up various technical and conceptual shortcomings in brain age approaches that inhibit their practical utility, but also provide a potential solution in the form of VNNs.** We are happy to answer any further concerns in this regard.
> > >
> > > **Reason for focusing on cortical thickness.**  Among the measures of thickness, volume, and area, cortical volume data is the most likely to be biased by the head size of individuals in the dataset, while cortical thickness is the least. Also, by limiting our analysis to only one modality (thickness in this case), the scope of our analysis was limited to 148 features. From a statistical perspective, this choice kept the ratio between the number of data samples available to the number of features to be the maximum possible while capturing the information across the cortex.
> > >
> > >
> > > $^1$ As observed in Appendix J, increasing the number of learnable parameters by about 10 times improves the chronological age prediction performance. However, these modifications take away our ability to comment on anatomical interpretability of VNNs and hence, do not add anything insightful to the task of brain age prediction.
> > >
> > > $^2$ Figure 7 in supplementary material adds to discussion in Section 4.1 in this regard.
> > >
> > > $^3$We have argued previously that post-hoc explainability is not conclusive or robust if shown to exist only on one instance.

---

> > > > ### Comment · Reviewer_mFGa · 2023-08-13
> > > >
> > > > I am very satisfied by the answers that the authors have been providing to my questions, and I thank them for the time taken. In general I am more convinced about the relevant contributions of this work and thus its contributions to the field. Even though we agree on the strengths of the paper and in the fact that there is a skewed focus on the performance of the brain age gap in literature, it seems we strongly disagree about the need for more baselines in the context of this same literature.
> > > >
> > > > I will increase the score I gave to this paper to 7 (accept), and increase the Soundness and Contribution scores to 4. I'm more confident about the relevance of this work compared to when I first reviewed it, and I hope the remaining reviewers can engage in this discussion too.

---

> > > > > ### Author Response · Authors · 2023-08-13
> > > > > **Thank you**
> > > > >
> > > > > Thank you for raising your recommendation for our paper. We greatly appreciate this discussion, which brought into focus many relevant aspects of the brain age prediction application.

---

### Author Rebuttal · Authors · 2023-08-07

We are grateful to all reviewers for their insightful feedback and appreciation of our work. We have provided individual responses to all reviewers. In this global response, we highlight the major advantages offered by VNNs over existing approaches and a summary of our response to two common concerns raised by the reviewers. *Also, the attached pdf file provides the results for cross-validation of brain age results on another public dataset on Alzheimer's disease (AD).*

To begin with, we highlight the following two *strengths of VNNs as a data analysis tool*.

- **Inherent explainability.** Most existing deep learning models offer explainability via post-hoc, model agnostic approaches. Such approaches are vulnerable to inconsistent results due to a range of factors and need extensive computation. In contrast, explainability is an *inherent feature* of VNN models and is straightforward to infer.
- **Scale-free characteristic of VNNs for cross-validation.** Because VNNs are scale-free and can process a dataset of arbitrary dimensionality, it is feasible to cross-validate the inference over datasets that may have a different number of features without any re-training or modifications to the VNN. This is a relevant feature for data analysis in neuroimaging where datasets describing the same phenomenon can have different dimensionalities. In the context of brain age, we leverage the scale-free property of VNNs to cross-validate anatomical interpretability associated with elevated brain age gap in the ADNI1 dataset (see **Fig. S2** in the attached pdf file), where we observed spatially consistent patterns associated with elevated brain age gap in AD across datasets of dimensions 100, 200, and 400 using VNNs that had been trained only on XYZ dataset (dimension:100; see supplementary file).

Next, we emphasize on the following *contribution of VNNs to the application of brain age prediction*.

- **Methodological clarity to brain age estimate.** Our results demonstrate that VNNs provide methodological clarity to all major aspects of brain age prediction. Specifically, VNNs learning 'something significant' from the chronological age data as regression models is a necessary first step for attaining robust anatomical interpretability (corroborated by additional results in Fig. 13 in supplementary material). Furthermore, the VNNs learn to exploit the eigenvectors of anatomical covariance matrix in a certain manner when trained on the task of chronological age prediction. Importantly, the anatomical interpretability offered by VNNs to the brain age gap hinges on exploitation of specific eigenvectors of the anatomical covariance matrix. Thus, elevated brain age gap in a population with adverse health condition can be linked to specific brain regions and the eigenvectors of the anatomical covariance matrix. Further, the analysis of regional residuals revealed that the utility of the age-bias correction step is limited to projecting the VNN output onto a space where a clinician can observe brain age relative to chronological age.


There were two common **concerns raised by the reviewers**. Our response to these concerns can be summarized as follows.


- **Lack of empirical evidence on additional data.** To address this, we have provided results on a dataset of 47 controls, 71 individuals with MCI, and 33 individuals with dementia from the standardized 3.0 T ADNI1 dataset (see [*] for details) from the wider ADNI database in the attached pdf file. We chose this standardized dataset in the spirit of reproducibility and to avoid selection bias. *Cross-validation using models trained on OASIS-3.* The results in **Fig. S1** were obtained from the models that were trained on OASIS-3 dataset. The observations on ADNI1 were highly consistent with those made in the paper. Specifically, in ADNI1 dataset, brain age gap was significantly higher for individuals with dementia as compared to healthy controls, with MCI in between them. Also, the subcallosal, entorhinal, temporal pole, and superior temporal regions were cross-validated on this cohort as being contributors to elevated brain age gap in the combined cohort of dementia and MCI (equivalent to AD+ group from OASIS-3).
- **Novelty, conceptual contributions, and comparisons with existing methods.** As highlighted above, (i) VNNs provide significant methodological clarity to major components of brain age prediction, and (ii) the explainability offered by VNNs is more robust and stable than that offered by existing approaches. We have responded to this concern, in part, in individual responses to all reviewers. The most extensive response has been provided in the comment titled ‘Contributions and novelty’ in response to the reviewer mFGa.

[*] Wyman BT et.al,  Standardization of analysis sets for reporting results from ADNI MRI data. Alzheimers Dement. 2013.

If our response has addressed your concerns, we will greatly appreciate it if you could re-evaluate your rating of the paper accordingly. We are happy to address any further concerns that may arise during the reviewer-author discussion period.


**Acknowledgement.** Data used in the preparation of the results for ADNI1 dataset were obtained from the Alzheimer’s Disease Neuroimaging Initiative (ADNI) database. The ADNI was launched in 2003 as a public-private partnership, led by Principal Investigator Michael W. Weiner, MD. The primary goal of ADNI has been to test whether serial magnetic resonance imaging (MRI), positron emission tomography (PET), other biological markers, and clinical and neuropsychological assessment can be combined to measure the progression of mild cognitive impairment (MCI) and early Alzheimer’s disease (AD).

---

> ### Author Response · Authors · 2023-08-18
> **Explanation-driven VNNs versus performance-driven baselines for brain age**
>
> We again thank the reviewers for their comments.  An unaddressed concern during the discussion period has been about insufficient comparisons with existing performance-driven baselines. To this end, we provide the following brief remark.
>
> The current state-of-the-art in brain age prediction is almost exclusively focused on performance-driven methods, i.e., their primary focus has been on how well the method can predict the chronological age for a healthy population. A key conceptual shortcoming of these methods is that chronological age is a noisy measure for brain age, even for a healthy population. From a methods' perspective, we have argued that these methods fail to provide robust validation of the anatomic plausibility of the elevated brain age gap in neurodegeneration. Such shortcomings have been identified before (*see references Cole et al, 2017, Bashyam et al, 2020 in the paper and Jirsaraie, et. al 2023 in the responses*).
>
> In contrast, VNNs provide an ​explanation-driven perspective to brain age prediction. Specifically, we primarily focus on what properties the VNNs gain when trained to predict chronological age of healthy controls. These properties are described in terms of exploiting eigenvectors of the anatomical covariance matrix by a VNN model. We also demonstrate that these properties enable the anatomical interpretability of brain age. Moreover, the resilience of the explanation offered by VNNs to brain age has been validated over 100 distinctly trained VNN models and across datasets. In principle, our paper has provided a novel perspective to brain age prediction, to which we find no convincing benchmarks for a meaningful comparison.
>
> Existing works have shown that machine learning models that leverage high-dimensional voxel level datasets or MRI images can achieve a much superior performance at predicting chronological age of healthy controls than that achieved by VNNs using the coarse ROI-level cortical thickness data. To clarify this, we will provide a short review of the performance achieved on OASIS-3 dataset by various existing machine learning methods and imaging modalities in the current literature. Also, we will mention that exploring the utility of VNNs in brain age prediction application with high dimensional voxel-level data is a potential direction to extend our work in the future.

---

### Decision · Program_Chairs · 2023-09-21

**Decision:**

Accept (poster)

**Comment:**

This submission generated much discussion. It was noted that the comparision to baselines is light. Other aspects of the manuscript were put forward, in particular in the direction of conceptual contributions to brain-age study as well as region-level contributions to brain age.